# HOXA9 promotes MYC-mediated leukemogenesis by maintaining gene expression for multiple anti-apoptotic pathways

Ryo Miyamoto[1], Akinori Kanai[2], Hiroshi Okuda[1], Yosuke Komata[1], Satoshi Takahashi[1,3], Hirotaka Matsui[4], Toshiya Inaba[2], Akihiko Yokoyama[1,5]*

[1]Tsuruoka Metabolomics Laboratory, National Cancer Center, Tsuruoka, Japan; [2]Department of Molecular Oncology and Leukemia Program Project, Research Institute for Radiation Biology and Medicine, Hiroshima University, Hiroshima, Japan; [3]Department of Hematology and Oncology, Kyoto University Graduate School of Medicine, Kyoto, Japan; [4]Department of Molecular Laboratory Medicine, Graduate School of Medical Sciences, Kumamoto University, Kumamoto, Japan; [5]Division of Hematological Malignancy, National Cancer Center Research Institute, Tokyo, Japan

**Abstract** HOXA9 is often highly expressed in leukemias. However, its precise roles in leukemogenesis remain elusive. Here, we show that HOXA9 maintains gene expression for multiple anti-apoptotic pathways to promote leukemogenesis. In MLL fusion-mediated leukemia, MLL fusion directly activates the expression of MYC and HOXA9. Combined expression of MYC and HOXA9 induced leukemia, whereas single gene transduction of either did not, indicating a synergy between MYC and HOXA9. HOXA9 sustained expression of the genes implicated in the hematopoietic precursor identity when expressed in hematopoietic precursors, but did not reactivate it once silenced. Among the HOXA9 target genes, *BCL2* and *SOX4* synergistically induced leukemia with *MYC*. Not only BCL2, but also SOX4 suppressed apoptosis, indicating that multiple anti-apoptotic pathways underlie cooperative leukemogenesis by HOXA9 and MYC. These results demonstrate that HOXA9 is a crucial transcriptional maintenance factor that promotes MYC-mediated leukemogenesis, potentially explaining why HOXA9 is highly expressed in many leukemias.

*For correspondence: ayokoyam@ncc-tmc.jp

## Introduction

Mutations of transcriptional regulators often cause aberrant gene regulation of hematopoietic cells, which leads to leukemia. Structural alterations of the mixed lineage leukemia gene (*KMT2A* also known as *MLL*) by chromosomal translocations cause malignant leukemia that often associates with poor prognosis despite the current intensive treatment regimens (*Tamai and Inokuchi, 2010*). *KMT2A* encodes a transcriptional regulator termed MLL that maintains segment-specific expression of homeobox (*HOX*) genes during embryogenesis (*Yu et al., 1998*), which determines the positional identity within the body (*Luo et al., 2019*; *Deschamps and van Nes, 2005*; *Wang et al., 2009*). During hematopoiesis, MLL also maintains the expression of posterior *HOXA* genes and *MEIS1* (another homeobox gene), which promote the expansion of hematopoietic stem cells and immature progenitors (*Jude et al., 2007*; *Krivtsov et al., 2006*; *McMahon et al., 2007*; *Thorsteinsdottir et al., 2002*; *Yagi et al., 1998*). The oncogenic MLL fusion protein constitutively activates its target genes by constitutively recruiting transcription initiation/elongation factors thereto (*Yokoyama et al., 2010*; *Lin et al., 2010*; *Okuda et al., 2015*). Consequently, *HOXA9* and *MEIS1* are highly transcribed in

MLL fusion-mediated leukemia (*Krivtsov et al., 2006*). Forced expression of HOXA9 (but not MEIS1) immortalize hematopoietic progenitor cells (HPCs) ex vivo (*Schnabel et al., 2000*; *Kroon et al., 1998*). Co-expression of HOXA9 with MEIS1 causes leukemia in mice which recapitulates MLL fusion-mediated leukemia (*Kroon et al., 1998*). Moreover, overexpression of HOXA9 is observed in many non-MLL fusion-mediated leukemias such as those with *NPM1* mutation and *NUP98* fusion and is associated with poor prognosis (*Collins and Hess, 2016*). These findings highlight HOXA9 as a major contributing factor in leukemogenesis. Nevertheless, the mechanism by which HOXA9 promotes oncogenesis remains elusive.

HOXA9 is considered to function as a transcription factor, which retains a sequence-specific DNA binding ability. HOX proteins have an evolutionarily conserved homeodomain which possesses strong sequence preferences (*Berger et al., 2008*). HOXA9 associates with other homeodomain proteins such as PBX and MEIS family proteins (*Schnabel et al., 2000*; *Shen et al., 1999*). HOXA9 and those HOXA9 cofactors form a stable complex on a DNA fragment harboring consensus sequences for each homeodomain protein (*Shen et al., 1999*; *Chang et al., 1996*), suggesting that they form a complex of different combinations in a locus-specific manner depending on the availability of the binding sites. Recently, it has been reported that HOXA9 specifically associates with enhancer apparatuses (e.g. MLL3/4) to regulate gene expression (*Sun et al., 2018*; *Huang et al., 2012*; *Zhong et al., 2018*). However, the mechanisms by which HOXA9 activates gene expression remain largely unclear.

In this study, we reveal the oncogenic roles for HOXA9 and its target gene products in leukemogenesis and its unique mode of function as a transcriptional maintenance factor that preserves an identity of a hematopoietic precursor.

## Results

### MLL fusion proteins and HOXA9 sustain MYC expression against differentiation-induced transcriptional suppression

To identify the direct target genes of MLL fusion proteins, we first examined the genome-wide localization pattern of MLL fusion proteins by chromatin immunoprecipitation (ChIP) followed by deep sequencing (ChIP-seq), using HB1119 cells, a cell line expressing the MLL-ENL fusion protein. We observed MLL ChIP signals on the *MYC*, *HOXA9*, *HOXA10*, and *MEIS1* loci (*Figure 1A*; *Okuda et al., 2017*), which were further confirmed by ChIP-quantitative polymerase chain reaction (qPCR) analysis (*Figure 1—figure supplement 1A*). These ChIP signals can be mostly attributed to MLL-ENL as the knockdown of wild-type MLL did not affect the MLL ChIP signals (*Figure 1A* and *Figure 1—figure supplement 1B,C*; *Okuda et al., 2017*). The distribution of MLL-ENL was enriched at transcription start sites in a genome-wide manner, which is similar to that of wild-type MLL observed in non-MLL fusion cell lines including HEK293T and REH (*Okuda et al., 2017*; *Miyamoto et al., 2020*).

To characterize the dynamic changes in MLL target gene expression during differentiation, we isolated bone marrow cells from mice at various differentiation stages ranging from the most immature population (c-Kit$^+$, Sca1$^+$, Lineage$^-$; KSL) containing hematopoietic stem cells to highly differentiated hematopoietic cells (c-Kit$^{low}$/Mac1$^{high}$) by fluorescence-activated cell sorting (FACS) and performed quantitative reverse transcription-polymerase chain reaction (qRT-PCR) analysis. For comparison, we analyzed leukemia cells (LCs) harvested from mice suffering with MLL-AF10-induced leukemia (MLL-AF10-LCs) and immortalized cells (ICs) transformed by MLL-ENL ex vivo (MLL-ENL-ICs). In accordance with previous reports (*Krivtsov et al., 2006*; *Somervaille and Cleary, 2006*; *Yokoyama et al., 2013*), the expression levels of *Hoxa9*, *Hoxa10*, and *Meis1* were downregulated during normal hematopoietic differentiation but remained high in MLL fusion-expressing cells (*Figure 1B*). *Myc* was highly expressed in KSL, common myeloid progenitors (CMPs), and granulocyte/macrophage progenitors (GMPs), which contain actively dividing populations (*Passegué et al., 2005*), but was completely suppressed at highly differentiated c-kit$^{low}$/Mac1$^{high}$ stages. In the two MLL fusion-expressing cell lines, *Myc* was expressed at comparable levels to those in the progenitor fractions (CMP and GMP; *Figure 1B*). *Mxd1*, a differentiation marker, was highly expressed in differentiated populations alone. These results indicate that *Myc*, *Hoxa9*, *Hoxa10*, and *Meis1* are

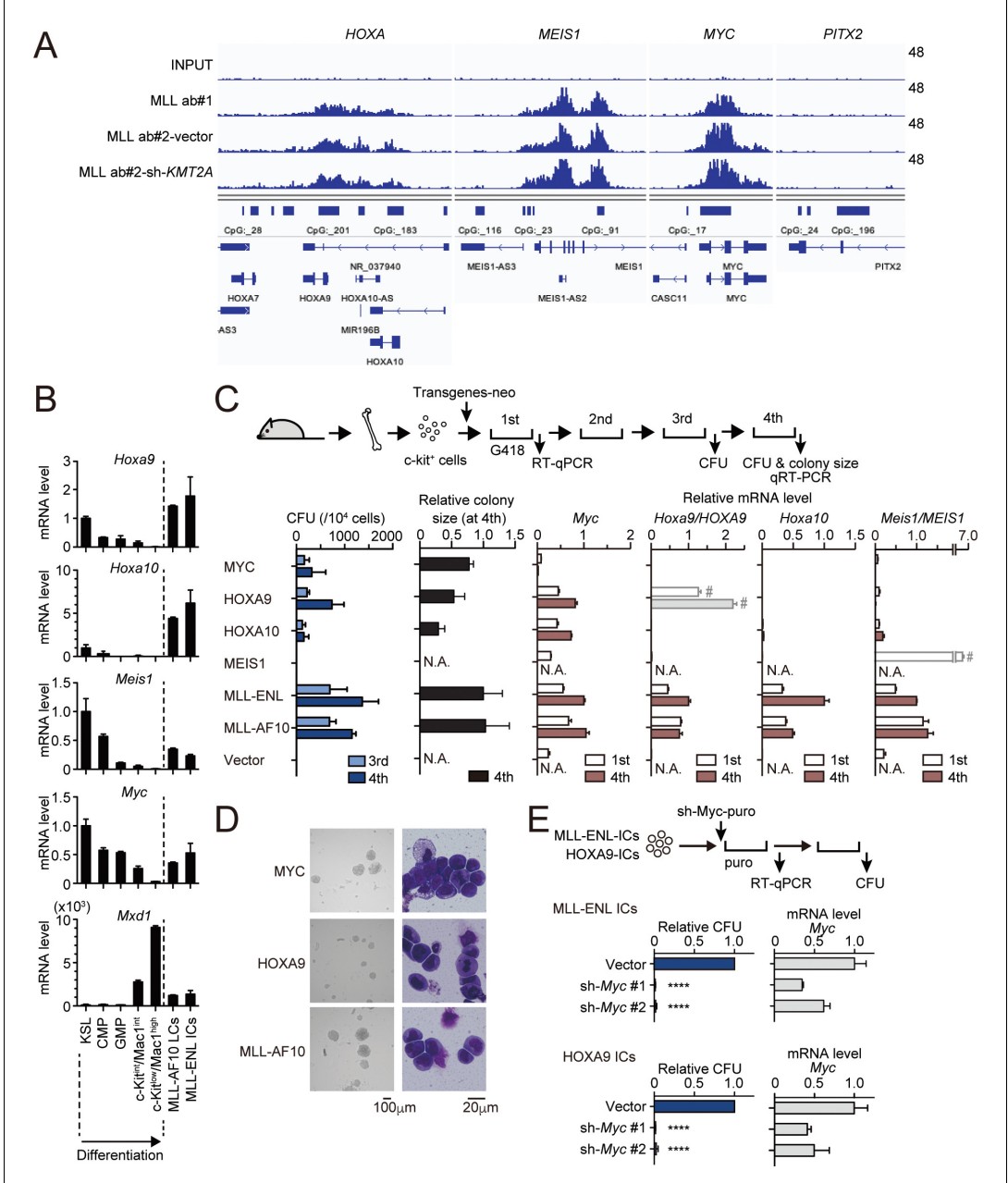

**Figure 1.** MLL fusion proteins and HOXA9 sustain MYC expression against differentiation-induced transcriptional suppression. (**A**) Genomic localization of MLL-ENL in HB1119 cells. ChIP signals at the loci of posterior *HOXA* genes, *MEIS1*, *MYC*, and *PITX2* (negative control) are shown using the Integrative Genomics Viewer (The Broad Institute). HB1119 cells were transduced with shRNA specific for wild-type (wt) MLL but not MLL-ENL (sh-*KMT2A*) to deplete wt MLL, as shown in ***Figure 1—figure supplement 1B***. Ab: antibody. (**B**) Expression of MLL-target genes and *Mxd1* (a differentiation marker) during myeloid differentiation. Bone marrow cells at various differentiation stages were obtained by FACS sorting and analyzed by qRT-PCR. Expression levels relative to KSL are shown (Mean with SD, n = 3, PCR replicates). MLL-AF10-LCs and MLL-ENL-ICs were included in the analysis for comparison. KSL: c-Kit[+], Sca1[+], and Lineage[−]; CMP: common myeloid progenitor; GMP: granulocyte macrophage progenitor; int: intermediate. (**C**) Transforming potential of MLL target genes. Clonogenic potential of the indicated constructs was analyzed by myeloid progenitor transformation assays. Colony forming unit per 10[4] cells (CFU) (Mean with SD, n = 3, biological replicates), relative colony size (Mean with SD, n ≥ 100), and relative mRNA levels of indicated genes (Mean with SD, n = 3, PCR replicates) were measured at the indicated time points. #: Both endogenous murine transcripts and exogenous human transcripts were detected by the qPCR primer set used and the columns were shown with faded color. N.A.: not assessed. (**D**) Morphologies of the colonies and transformed cells. Representative images of bright field (left) and May-Grunwald-Giemsa staining (right) are shown with scale bars. (**E**) Effects of *Myc* knockdown on MLL-ENL- and HOXA9-ICs. Relative CFU (Mean with SD, n = 3, biological replicates) and mRNA level of *Myc* (Mean with SD, n = 3 PCR replicates) are shown. Statistical analysis was performed using ordinary one-way ANOVA with the vector control. ****p < 0.0001.

*Figure 1 continued on next page*

*Figure 1 continued*

The online version of this article includes the following figure supplement(s) for figure 1:

**Figure supplement 1.** Association of the MLL-ENL complex on target promoters.
**Figure supplement 2.** Expression of the MYC protein after *Myc* knockdown.

intrinsically programmed to be silenced in normal hematopoietic differentiation but are aberrantly maintained by MLL fusion proteins.

To assess the oncogenic potential of MLL target genes, we performed myeloid progenitor transformation assays (*Lavau et al., 1997*; *Okuda and Yokoyama, 2017a*), wherein HPCs were isolated from mice, retrovirally transduced with each MLL target gene, and cultured in a semi-solid medium supplemented with cytokines promoting myeloid lineage differentiation (*Figure 1C,D*). HPCs exogenously expressing MLL-ENL or MLL-AF10 produced a large number of colonies in the third and fourth passages with high mRNA levels of *Myc*, *Hoxa9*, *Hoxa10*, and *Meis1* in colonies of the first passage, confirming their potent transforming capacities. These cells were considered 'immortalized' as they proliferate indefinitely in this ex vivo culture (*Lavau et al., 1997*; *Okuda and Yokoyama, 2017a*). Ectopic expression of MYC immortalized HPCs; therefore, MYC expression is sufficient to induce proliferation of HPCs. Endogenous *Myc* expression was severely diminished in MYC-ICs, demonstrating that *Myc* expression is programed to be silenced and cannot be sustained by MYC itself. Ectopic expression of HOXA9 and HOXA10 (but not MEIS1) immortalized HPCs. Endogenous *Myc* expression was maintained in HOXA9/A10-ICs, indicating that HOXA9/A10 can sustain *Myc* expression against differentiation-induced transcriptional suppression. It should be noted that the colony size of HOXA9/A10-ICs was relatively small compared with MYC- or MLL fusion-ICs, suggesting a weaker proliferative potential. Knockdown of *Myc* by shRNA completely repressed the colony-forming ability of MLL-ENL-ICs and HOXA9-ICs (*Figure 1E* and *Figure 1—figure supplement 2*). Taken together, these results demonstrate that MLL fusion proteins and HOXA9 maintain *Myc* expression despite of the differentiation-induced transcriptional suppression, and the maintenance of MYC expression is indispensable for immortalization of HPCs ex vivo.

## HOXA9 confers the identity of a hematopoietic precursor while MYC drives anabolic pathways

To identify the genes specifically regulated by HOXA9 but not by MYC, we performed RNA-seq analysis of HOXA9-ICs and MYC-ICs which do not express *HOXA9* (*Figure 1C*). Genes highly expressed in HOXA9-ICs but lowly expressed in MYC-ICs (defined as 'HOXA9 high signature') were associated with hematopoietic identity/functions. In contrast, genes highly expressed in MYC-ICs and lowly expressed in HOXA9-ICs (defined as 'MYC high signature') were associated with anabolic pathways involved in nucleotide/protein production (*Figure 2A,B*). Pathways involved in lipid metabolism were commonly upregulated in both HOXA9- and MYC-ICs compared with non-immortalized c-kit-positive HPCs (*Figure 2—figure supplement 1A,B*). MLL-AF10-ICs, which express endogenous *Hoxa9* and *Myc* at high levels (*Figure 1C*), expressed both HOXA9 high and MYC high signature genes (*Figure 2A–C*), indicating that MLL-AF10-ICs possess MYC-mediated highly proliferative potential and HOXA9-mediated hematopoietic identity. These differences between the HOXA9 high and MYC high signatures indicate that HOXA9 maintains the identity of a hematopoietic precursor, while MYC promotes proliferation by upregulating anabolic pathways.

## Apoptosis is induced by MYC, and alleviated by HOXA9 and MLL-AF10

Given that excessive MYC activity promotes apoptosis (*McMahon, 2014*), we evaluated the apoptotic tendencies in immortalized HPCs. MYC-ICs exhibited increased γH2AX, cleaved poly (ADP) ribose polymerase (PARP), and cleaved caspase 3 levels, indicative of a high degree of replication stress and apoptosis (*Figure 3A*). FACS analysis with Annexin V also showed that MYC-ICs had more significant apoptotic fraction than that of HOXA9-ICs (*Figure 3B*). MLL-AF10-ICs exhibited weak apoptotic tendencies similarly to HOXA9-ICs, although their MYC expression tended to be higher than HOXA9-ICs (*Figures 1C* and *3A*). Furthermore, most MYC-ICs underwent massive apoptosis 1 day after cytokine removal, whereas HOXA9-ICs and MLL-AF10-ICs showed relative resistance (*Figure 3C*) and exhibited successful recovery of the live cell population after cytokine reintroduction

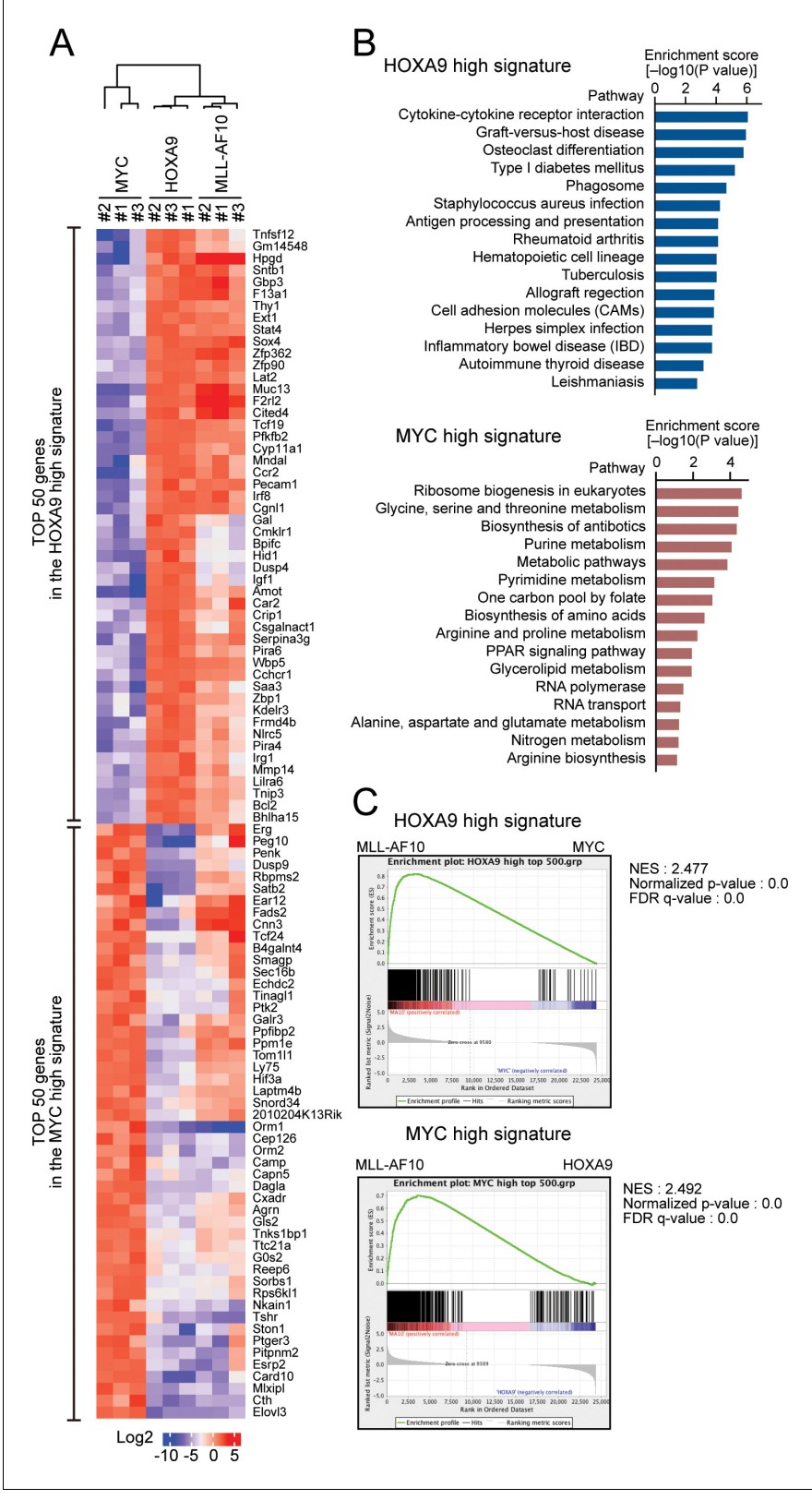

**Figure 2.** HOXA9 confers the identity of a hematopoietic precursor while MYC drives anabolic pathways. (**A**) Relative expression of the top 50 genes categorized as the HOXA9 high signature and the MYC high signature in HOXA9-, MYC-, and MLL-AF10-ICs. The RPKM data are provided in *Figure 2—source data 1*. (**B**) Pathways related to the HOXA9 high signature (blue) and the MYC high signature (red). The top 500 genes in the HOXA9

*Figure 2 continued on next page*

*Figure 2 continued*

high or MYC high signatures were subjected to KEGG pathway analysis. The summary is provided in *Figure 2—source data 1* (C) Gene set enrichment analysis of the HOXA9 high and MYC high signatures in MLL-AF10-ICs compared with MYC-ICs (top) and HOXA9-ICs (bottom), respectively.

The online version of this article includes the following source data and figure supplement(s) for figure 2:

**Source data 1.** Gene expression profiles of HOXA9-, MYC-, and MLL-AF10-transformed cells.

**Figure supplement 1.** Biological pathways commonly regulated by HOXA9 and MYC.

---

(*Figure 3—figure supplement 1*). These results indicate that HOXA9 confers anti-apoptotic properties.

Next, we evaluated the leukemogenic potential of HOXA9 and MYC in vivo. Transplantation of HPCs exogenously expressing MLL-AF10 into syngeneic mice induced leukemia with full penetrance (*Figure 3D*). In contrast, neither *MYC* nor its homologue *MYCN* could initiate leukemia within 200 days under these experimental conditions, indicating that activation of the MYC high signature alone

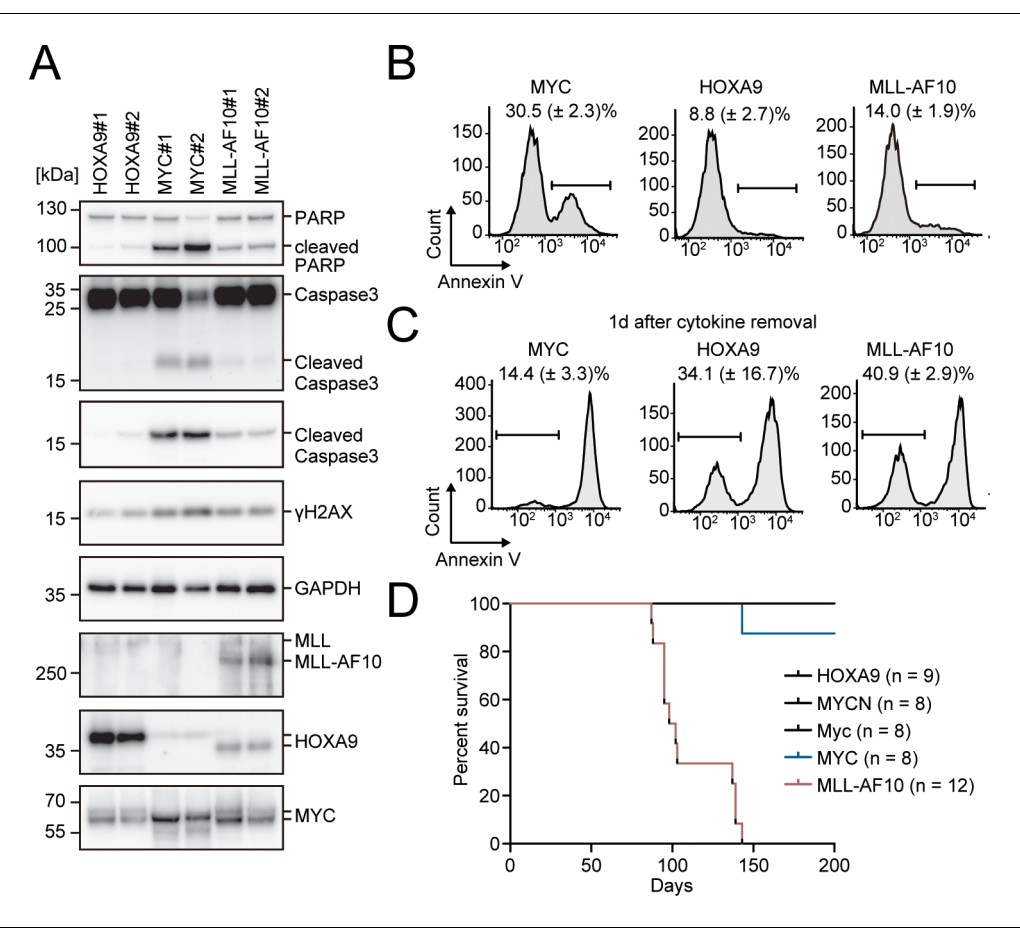

**Figure 3.** Apoptosis is induced by MYC, while is alleviated by HOXA9 and MLL-AF10. (**A**) Protein expression of transgenes and apoptotic markers in HOXA9-, MYC-, and MLL-AF10-ICs. (**B** and **C**) Apoptotic tendencies of HOXA9-, MYC-, and MLL-AF10-ICs. Representative FACS plots and the summarized data (Mean with SD, n = 3, biological replicates) of Annexin V staining of HOXA9-, MYC-, and MLL-AF10-ICs in the presence of cytokines (**B**) and 1d after their removal (**C**) are shown. (**D**) In vivo leukemogenic potential of MLL target genes and MLL-AF10. Kaplan-Meier curves of mice transplanted with HPCs transduced with the indicated genes and the number of replicates are shown.

The online version of this article includes the following figure supplement(s) for figure 3:

**Figure supplement 1.** | Survival of HOXA9-expressing cells upon cytokine withdrawal.

is insufficient for leukemogenesis in vivo. Although one recipient mouse of *MYC*-transduced HPCs became sick and was sacrificed approximately 150 days after transplantation, it did not exhibit any leukemia-associated signs. The mouse *Myc* gene also failed to induce leukemia. Thus, it is unlikely that the inability of MYC alone to induce leukemia under these experimental conditions is due to immune suppression of the cells expressing human transgenes. *HOXA9* did not induce leukemia within 200 days either, suggesting that the HOXA9 high signature alone is also insufficient to induce leukemia. Taken together, these results suggest that both high MYC activity and HOXA9-mediated resistance to apoptosis are necessary for driving leukemogenesis in vivo.

## HOXA9 promotes MYC-mediated leukemogenesis

We next examined the gene expression of patients with leukemia using publicly available microarray data of the Microarray Innovations in LEukemia (MILE) study (*Haferlach et al., 2010*). Of 108 cases with MLL translocation, 38 were acute myelogenous leukemia [AML], and 70 were acute lymphoblastic leukemia [ALL]. Most of the cases (78.7%) were categorized in the HOXA9$^{high}$/MEIS1$^{high}$ group, while some were in the HOXA9$^{low}$/MEIS1$^{high}$ (13.0%) or HOXA9$^{high}$/MEIS1$^{low}$ (8.3%) group (*Figure 4A*). *MYC* was expressed at high levels irrespective of *HOXA9* or *MEIS1* expression. These data indicate variability in transcriptional profiles among MLL fusion-mediated leukemia cases, with *MYC* expression remaining consistently high. HOXA9$^{low}$/MEIS1$^{high}$ leukemia was predominantly found in ALL, likely due to the MLL-AF4 cases, some of which do not express *HOXA* genes (*Lin et al., 2016*). HOXA9$^{high}$/MEIS1$^{low}$ leukemia was mainly found in AML (*Figure 4B*).

Next, we assessed the effects of the combined expression of MYC, HOXA9, and MEIS1 in mouse leukemia models. The MYC/HOXA9 combination exhibited high clonogenicity ex vivo, with colony-forming capacity and colony size comparable to those of MLL-ENL- and HOXA9/MEIS1-ICs (*Figure 4C,D*). A weak synergy between MYC and MEIS1 was also observed. HOXA9/MEIS1-ICs showed high *Myc* expression (*Figure 4—figure supplement 1A*) and MYC-dependent proliferation (*Figure 4—figure supplement 1B*), indicating that MYC is essential in HOXA9/MEIS1-mediated leukemic transformation. In the in vivo leukemogenesis assays, MYC, HOXA9, or MEIS1 expression alone did not initiate leukemia, whereas co-expression of HOXA9 and MEIS1 did in all recipient mice as previously reported (*Figure 4E*; *Kroon et al., 1998*). The combined expression of MYC/HOXA9 or MYC/MEIS1 induced leukemia in 38% and 18% of recipient mice, respectively, indicating a synergy of MYC with HOXA9 and MEIS1. qRT-PCR analysis showed that LCs maintained the expression patterns of endogenous *Hoxa9*, *Meis1*, and *Myc* similar to those of the respective ICs (*Figure 4—figure supplement 2*). We observed rapid onset of leukemia with full penetrance in the secondary transplantation for the three combinations tested, confirming the presence of leukemia-initiating cells (*Figure 4E*). These results indicate a cooperative role for HOXA9 and MEIS1 in MYC-mediated leukemogenesis with stronger synergy between HOXA9 and MYC.

All the MYC/HOXA9- and HOXA9/MEIS1-induced leukemias were positive to a myeloid marker Mac1 but negative to lymphoid marker B220 or CD3e (*Figure 4—figure supplement 3A,B*), indicating that both MYC/HOXA9- and HOXA9/MEIS1-combinations induced myeloid leukemia. Gene set enrichment analysis (GSEA) of MYC/HOXA9-LCs and HOXA9/MEIS1-LCs revealed that genes involved in protein synthesis (e.g. ribosome biogenesis, tRNA metabolic process) were enriched in MYC/HOXA9-LCs (*Figure 4—figure supplement 4A*), while tumour necrosis factor α signaling and KRAS pathways were enriched in HOXA9/MEIS1-LCs (*Figure 4—figure supplement 4B*). These results suggested that these two combinations promoted leukemogenesis in slightly different ways, wherein MYC-mediated anabolic pathways played more prominent roles in leukemogenesis by MYC/HOXA9 than by HOXA9/MEIS1.

## HOXA9 functions as a transcription maintenance factor

Some HOXA9 high signature genes, namely *Bcl2*, *Sox4*, and *Igf1*, have been implicated in leukemogenesis (*Zhang et al., 2013*; *Du et al., 2005*; *Delbridge et al., 2016*; *Steger et al., 2015*), suggesting that they may be responsible for the synergy between HOXA9 and MYC. Indeed, *Bcl2*, *Sox4*, and *Igf1* were highly transcribed in HPCs transformed by the combinations containing HOXA9 (i.e. MYC/HOXA9-ICs, HOXA9/vector-ICs, and HOXA9/MEIS1-ICs) but not those lacking it (i.e. MYC/vector-ICs and MYC/MEIS1-ICs) (*Figure 5A* and *Figure 5—figure supplement 1A*). Conditional loss of function experiments using HOXA9 conjugated with estrogen receptor (HOXA9-ER) confirmed

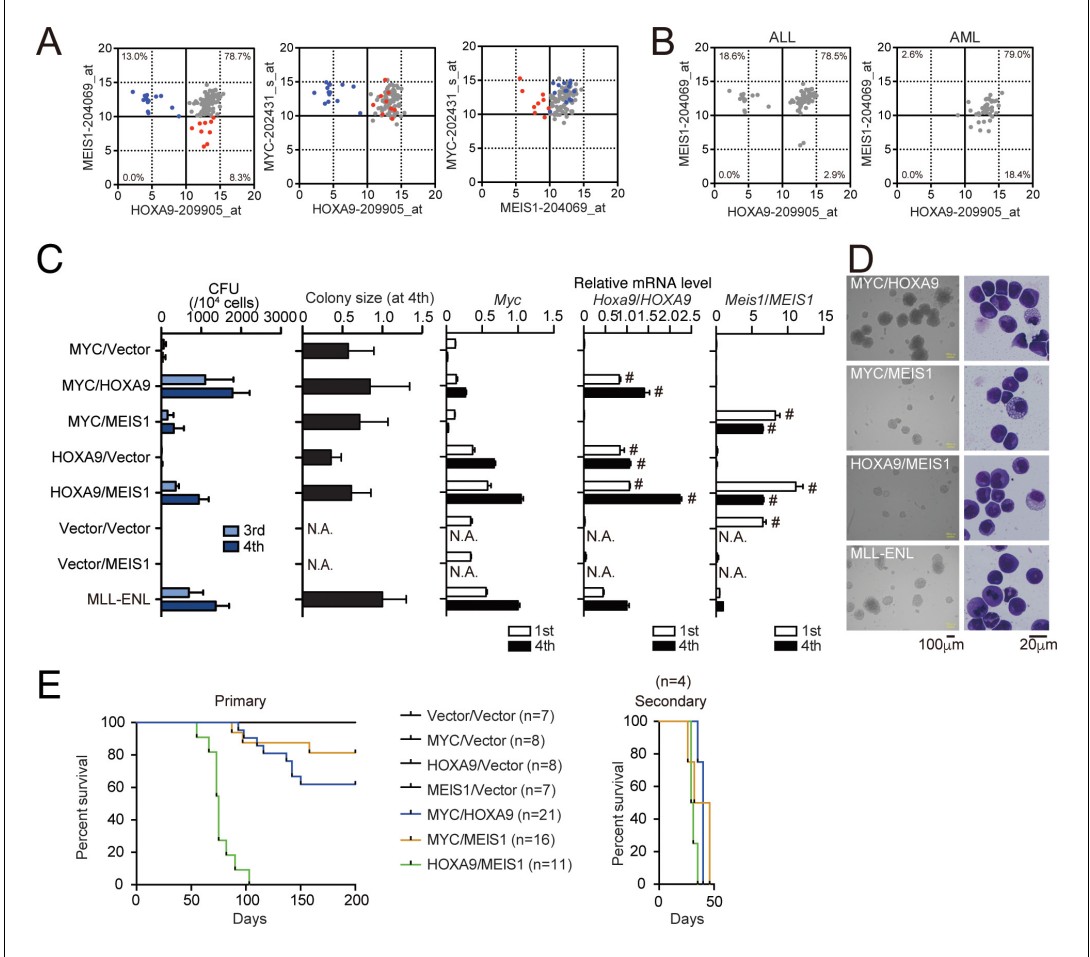

**Figure 4.** HOXA9 promotes MYC-mediated leukemogenesis. (**A** and **B**) Expression profiles of MLL fusion-mediated leukemia patients reported in the MILE study (*Haferlach et al., 2010*). Probe intensities of the indicated genes are plotted for all MLL fusion-mediated leukemia patients (**A**). Patients in the HOXA9$^{low}$/MEIS1$^{high}$ (blue) and HOXA9$^{high}$/MEIS1$^{low}$ (red) groups are highlighted. Probe intensities of *HOXA9* and *MEIS1* are plotted separately by leukemia phenotype (ALL or AML) (**B**). The expression profiles are provided in *Figure 4—source data 1*. (**C**) Transforming potential of various combinations of MLL target genes. CFU (Mean with SD, n = 3, biological replicates) and relative colony size (Mean with SD, n ≥ 100) are shown as in *Figure 1C*. (**D**) Morphologies of the colonies and transformed cells. Bright-field (left) and May-Grunwald-Giemsa staining (right) images are shown with scale bars. (**E**) In vivo leukemogenic potential of various oncogene combinations. Kaplan-Meier curves of mice transplanted with HPCs transduced with the indicated genes are shown as in *Figure 3D*. Bone marrow cells from moribund mice were harvested and used for secondary transplantation.

The online version of this article includes the following source data and figure supplement(s) for figure 4:

**Source data 1.** Gene expression profiles of MLL fusion-mediated leukemia patients in the MILE data.

**Figure supplement 1.** Role of MYC in HOXA9/MEIS1-mediated transformation.

**Figure supplement 2.** Expression of endogenous *Myc* and the transgenes in immortalized- and leukemia-cells.

**Figure supplement 3.** Phenotypes of MLL-AF10-, HOXA9/MEIS1-, and HOXA9/MYC-mediated leukemia cells at the endpoint.

**Figure supplement 4.** Biological pathways differentially regulated in HOXA9/MYC- and HOXA9/MEIS1-driven leukemias.

critical regulation of these genes by HOXA9 (*Figure 5B*). However, stepwise transduction of *MYC* followed by *HOXA9* failed to upregulate these HOXA9 target genes, indicating that HOXA9 could not reactivate its target genes once silenced (*Figure 5C* and *Figure 5—figure supplement 1B*). Conditional inactivation and subsequent reactivation of HOXA9 within the same cell population confirmed that HOXA9 cannot reactivate its target genes once silenced (*Figure 5—figure supplement 1C*). RNA-seq analysis of various human cell lines showed that *HOXA9* was highly expressed in all the MLL fusion cell lines tested (*Figure 5—figure supplement 2A*, highlighted in blue). *BCL2* and *SOX4* were also expressed but at different levels depending on the cell lines. In accord, most of the MLL fusion-mediated leukemia patient samples in the MILE study with high *HOXA9* expression

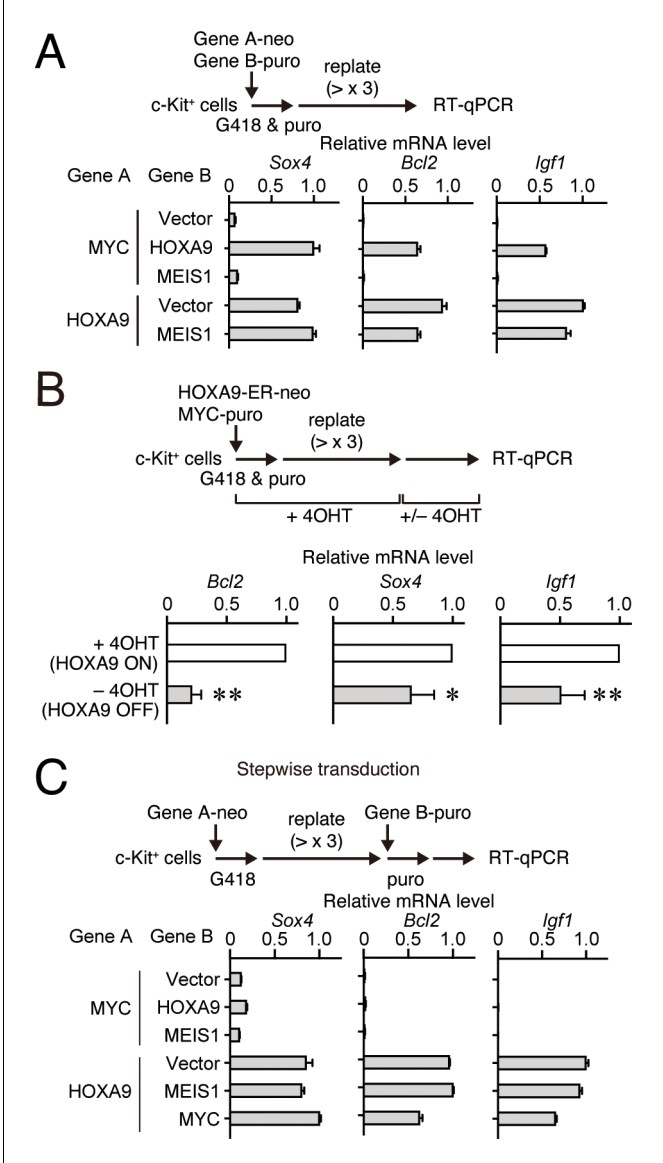

**Figure 5.** HOXA9 functions as a transcription maintenance factor. (**A**) Gene expression of HPCs immortalized by various transgenes. Relative mRNA levels of HOXA9 target genes (Mean with SD, n = 3, PCR replicates) in myeloid progenitors transformed by various combinations of MLL target genes are shown. Two genes were transduced into HPCs in a simultaneous manner. (**B**) Gene expression after inactivation of HOXA9. HOXA9-ER and MYC were doubly transduced into HPCs and cultured in the presence of 4-OHT ex vivo. After 4-OHT withdrawal, qRT-PCR analysis was performed for the indicated genes (Mean, n=4, biological replicates). Statistical analysis was performed using unpaired two-tailed Student's *t*-test. \*\*p< 0.01, \*p < 0.05. (**C**) Gene expression of HPCs immortalized by step-wise transduction of various transgenes. Relative mRNA levels of HOXA9 target genes in myeloid progenitors transformed by various combinations of MLL target genes are shown as in A. Two genes were transduced into HPCs in a stepwise manner.

The online version of this article includes the following figure supplement(s) for figure 5:

**Figure supplement 1.** HOXA9-dependent gene expression of various HOXA9 target genes.
**Figure supplement 2.** Gene expression of various HOXA9 target genes in cell lines and patients.

---

expressed *BCL2* and *SOX4* at high levels (*Figure 5—figure supplement 2B*; *Haferlach et al., 2010*). *IGF1* was only expressed in MV4-11 cells despite comparable *HOXA9* expression among the MLL fusion cell lines, supporting the notion that HOXA9 expression itself cannot trigger the expression of silenced HOXA9 target genes. These results indicate that HOXA9 is a transcriptional maintenance

factor that may be involved in the maintenance of chromatin structure previously activated by other transcriptional/epigenetic factors.

## BCL2 and SOX4 promote MYC-mediated leukemogenesis by alleviating apoptosis

To identify the roles for BCL2 and SOX4 in leukemic transformation, we evaluated the leukemogenic potential of the combined expression of BCL2 or SOX4 with MYC. In myeloid progenitor transformation assays, SOX4 by itself showed weak immortalization capacity as previously reported (*Zhang et al., 2013*), while BCL2 did not transform HPCs (*Figure 6A*). Co-expression of BCL2 or SOX4 with MYC led to a substantial increase in colony-forming capacity. The combined expression of BCL2 with MYC induced leukemia in vivo with a penetrance similar to that with the MYC/HOXA9 combination (*Figures 4E* and *6B*, and *Figure 6—figure supplement 1A*), in accord with previous reports (*Beverly and Varmus, 2009*; *Luo et al., 2005*). SOX4 also promoted MYC-mediated leukemogenesis in vivo (*Figure 6B* and *Figure 6—figure supplement 1A*), while neither MYC, BCL2, nor SOX4 alone induced leukemia within 200 days. FACS analysis indicated that all of MYC/SOX4-induced leukemias were of myeloid lineage (*Figure 6—figure supplement 1B*). In contrast, all MYC/BCL2-induced leukemias were of lymphoid lineage, the half of which was B-cell type, and the other half was T-cell type, consistent with a previous report (*Luo et al., 2005*). It should be noted that single gene transduction of MYC and SOX4 induced leukemia by others in different settings, albeit with low penetrance (*Du et al., 2005*; *Luo et al., 2005*). These differences are possibly due to the differences of virus titers and viral genome integration-related gene activation. Furthermore, the combined expression of HOXA9, BCL2, or SOX4 with MYC alleviated apoptotic tendencies (*Figure 6C,D*). Although SOX4 is reported to modulate transcription of pro/anti-apoptotic genes (*Ramezani-Rad et al., 2013*), their expression was not drastically altered by SOX4 in this context (*Figure 6E,F*). Thus, the mechanism underlying the anti-apoptotic properties of SOX4 in this setting is currently unclear. Taken together, the results indicate that multiple anti-apoptotic pathways mediated by BCL2 and SOX4 promote MYC-mediated leukemic transformation.

## Endogenous BCL2 and SOX4 support the initiation and maintenance of leukemia

To examine the roles for endogenous BCL2 and SOX4 in the initiation and maintenance of LCs, we conducted myeloid progenitor transformation and in vivo leukemogenesis assays using *Bcl2*- and *Sox4*-knockout HPCs. We transduced various oncogenes into HPCs isolated from fetal livers of *Bcl2*- and *Sox4*-knockout embryos (*Kamada et al., 1995*; *Schilham et al., 1996*) and cultured them ex vivo. Neither *Bcl2* nor *Sox4* deletion affected the immortalization of HPCs by HOXA9, MYC, or MLL-AF10, indicating that BCL2 and SOX4 are dispensable for proliferation ex vivo (*Figure 7—figure supplement 1*). On the other hand, *Bcl2* deletion delayed the onset of leukemia induced by MLL-AF10 and the HOXA9/MEIS1 combination in vivo (*Figure 7A* and *Figure 7—figure supplement 2A*). *Sox4* deletion also delayed the onset of HOXA9/MEIS1-induced leukemia, although it did not affect MLL-AF10-induced leukemia (*Figure 7B* and *Figure 7—figure supplement 2A*). We also analyzed the steady-state apoptosis of LCs harvested from moribund mice with apoptotic markers (cleaved caspase, γH2AX and Annexin V). HOXA9/MEIS1-LCs were slightly more apoptotic in the absence of *Bcl2* and *Sox4* (*Figure 7—figure supplement 2B,C*). However, we did not observe substantial differences in MLL-AF10-LCs between the WT and *Bcl2*/*Sox4* KO genotypes. These results are consistent with the different kinetics of leukemia onset (*Figure 7A,B*), and the minor differences in apoptotic tendencies in full-blown leukemia cells suggest that these LCs had acquired adequate anti-apoptotic properties at the time of disease presentation. These results indicate that endogenous BCL2 and SOX4 partially contribute to initiating leukemia in vivo.

Next, we examined the roles for endogenous BCL2 and SOX4 in maintaining leukemia-initiating cells using a mouse leukemia cell line that we previously established with MLL-ENL (*Okuda et al., 2017*). CRISPR/Cas9-mediated sgRNA competition assays indicated the negligible contribution of *Bcl2* and *Sox4* to MLL-ENL-LC proliferation ex vivo (*Figure 7—figure supplement 3*). To test their roles in vivo, we transplanted *Bcl2*- or *Sox4*-deficient LCs into syngeneic mice, where we observed delayed onset and reduced incidence rate for both *Bcl2*- and *Sox4*-knockout LCs (*Figure 7C*). Forced expression of sgRNA-resistant cDNAs of *BCL2* and *SOX4* restored leukemogenic potential,

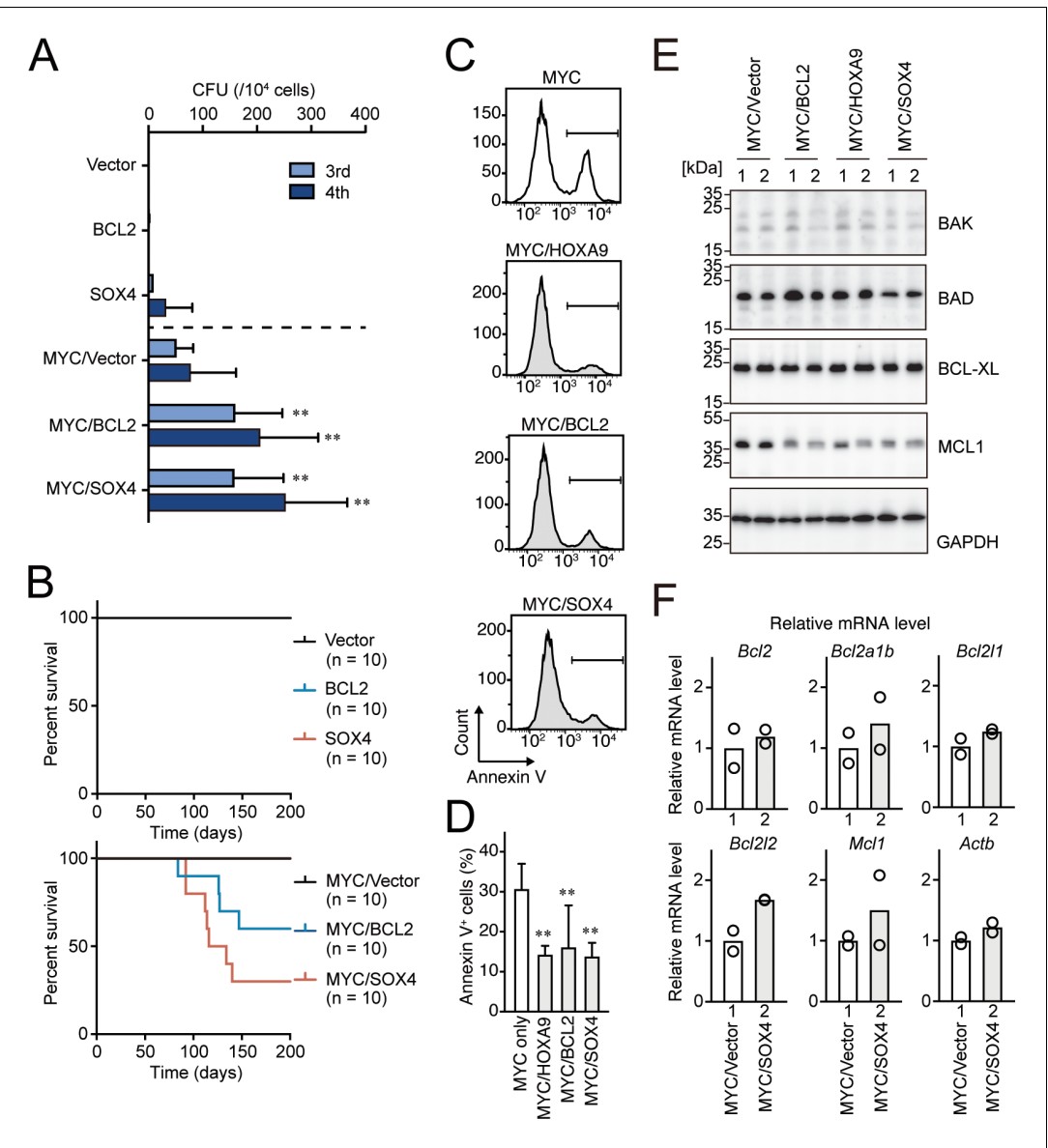

**Figure 6.** BCL2 and SOX4 promote MYC-mediated leukemogenesis by alleviating apoptosis. (**A**) Transforming potential of various combinations of MYC and HOXA9 target genes. CFU (Mean with SD, n = 3, biological replicates) is shown as in *Figure 1C*. (**B**) In vivo leukemogenic potential of various combinations of MYC and HOXA9 target genes. Kaplan-Meier curves of mice transplanted with HPCs transduced with the indicated genes are shown as in *Figure 3D*. (**C** and **D**) Apoptotic tendencies of *MYC*-expressing progenitors co-transduced with *HOXA9*, *BCL2*, or *SOX4*. Representative FACS plots (**C**) and the summarized data (**D**) (Mean with SD, n = 3, biological replicates) of Annexin V staining are shown. Statistical analysis was performed using ordinary one-way ANOVA with MYC-ICs. (**E**) Expression of apoptosis-related proteins in HPCs transformed by various combinations of transgenes. Western blots of HPCs transformed by indicated transgenes are shown (**F**) Relative expression levels of apoptosis-related genes in MYC/SOX4-ICs and MYC/vector-ICs. Relative mRNA levels of the indicated apoptosis-related genes (Mean, n = 2, biological replicates) are shown.

The online version of this article includes the following figure supplement(s) for figure 6:

**Figure supplement 1.** Phenotypes of MYC/BCL2- and MYC/SOX4-mediated leukemias at the endpoint.

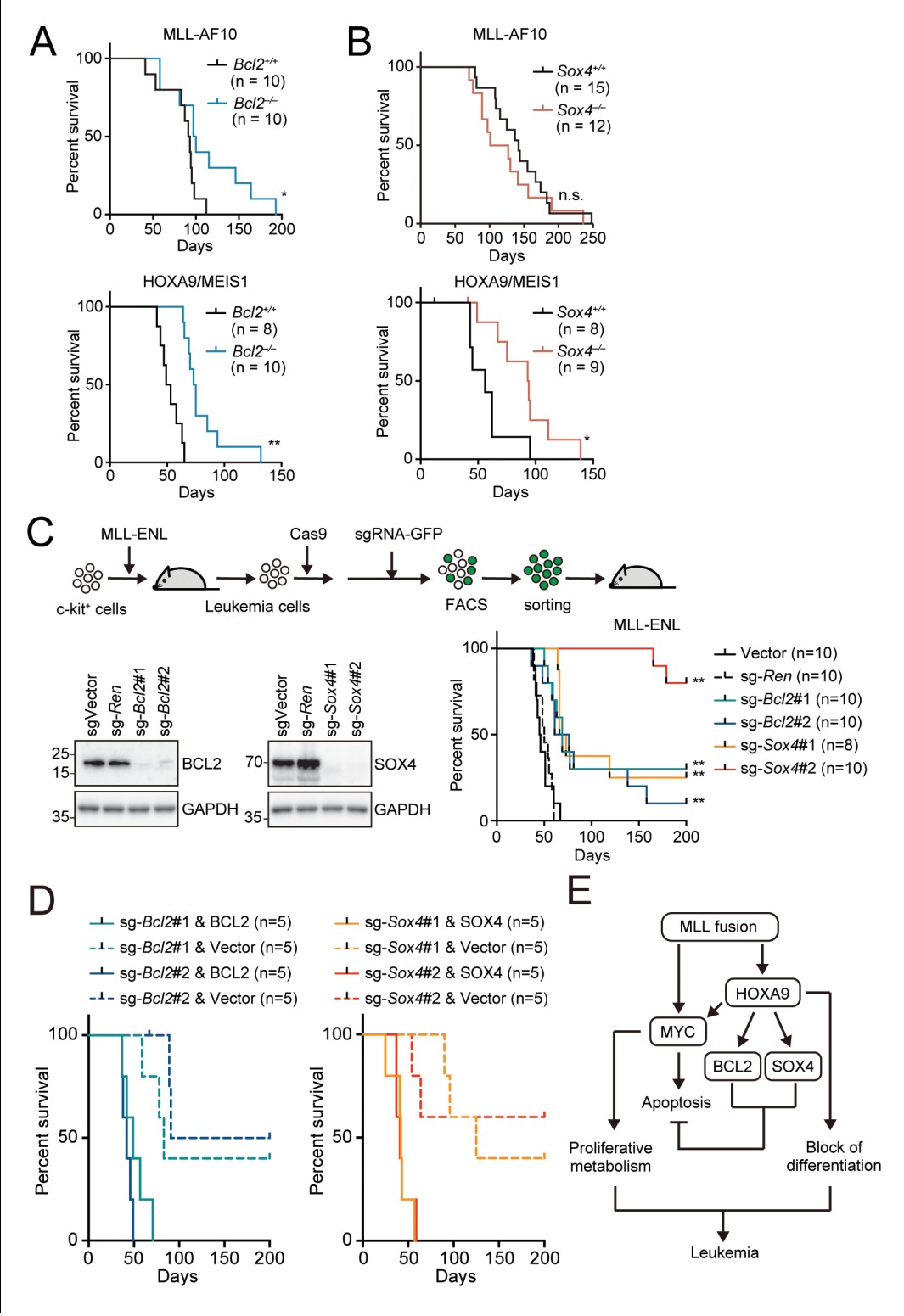

**Figure 7.** Endogenous BCL2 and SOX4 support the initiation and maintenance of leukemia. (**A**) Effects of *Bcl2*-deficiency on the initiation of leukemogenesis in vivo. HPCs were isolated from *Bcl2*$^{+/+}$ or *Bcl2*$^{-/-}$ embryos and transduced with the retroviruses for MLL-AF10 or the HOXA9/MEIS1 combination. Kaplan-Meier curves of mice transplanted with the transduced HPCs are shown. Statistical analysis was performed using the log-rank test and Bonferroni correction with the wildtype control. *p ≤ 0.05. (**B**) Effects of *Sox4*-deficiency on the initiation of leukemogenesis in vivo. In vivo leukemogenesis assay was performed on *Sox4*$^{+/+}$ or *Sox4*$^{-/-}$ embryos as in A (**C**)
*Figure 7 continued on next page*

*Figure 7 continued*

Effects of *Bcl2*- or *Sox4*-deficiency on the maintenance of leukemia initiating cells. Western blots of MLL-ENL leukemia cells transduced with sgRNA for BCL2 or SOX4 are shown on the left. Kaplan-Meier curves of mice transplanted with MLL-ENL leukemia cells transduced with the indicated sgRNAs are shown on the right. Statistical analysis was performed using the log-rank test and Bonferroni correction with the vector control. (D) Rescue of in vivo leukemogenic potential by sgRNA-resistant transgenes. Before transduction of sgRNA, MLL-ENL-LCs were transduced with sgRNA-resistant *BCL2* or *SOX4*. In vivo leukemogenesis assay was performed as described in *Figure 3D*. (E) A model illustrating HOXA9-mediated pathogenesis in MLL fusion-mediated leukemia.

The online version of this article includes the following figure supplement(s) for figure 7:

**Figure supplement 1.** Requirement of BCL2 and SOX4 in transformation ex vivo.
**Figure supplement 2.** Apoptotic tendencies of Bcl2- or Sox4-deficient leukemia cells.
**Figure supplement 3.** Effects of *Bcl2* or *Sox4* knockout on MLL-ENL-leukemia cells.

---

validating the on-target effects of sgRNAs (*Figure 7D*). These results suggest that MLL-ENL-LCs are partially dependent on endogenous BCL2 and SOX4 in vivo. Taken together, our results indicate that MLL fusion-mediated LCs depend on the expression of multiple anti-apoptotic genes via HOXA9 to varying degrees to achieve survival advantages for disease initiation and maintenance (*Figure 7E*).

## Discussion

In this study, we found that HOXA9 regulates a variety of genes to maintain hematopoietic precursor identity and its associated anti-apoptotic properties. In leukemic transformation, MLL fusion proteins exploit both HOXA9 and MYC downstream pathways. Accordingly, HOXA9 and MYC synergistically induce leukemia in mouse models. Thus, we propose that MLL fusion proteins employ two arms to promote oncogenesis: MYC-mediated proliferation and HOXA9-mediated resistance to differentiation/apoptosis.

It is widely accepted that two types of mutations need to occur before leukemia onset; class I mutations that confer proliferative advantages and class II mutations that block differentiation (*Gilliland, 2002*). However, it was unclear how MLL mutations fit this theory because MLL fusion-mediated leukemia does not require additional mutations in many cases (*Andersson et al., 2015*). A comparison of gene expression profiles of HOXA9- and MYC-transformed HPCs demonstrated that HOXA9 maintains the expression of a wide range of genes associated with hematopoietic precursor identity. Thus, HOXA9 appears to maintain the intrinsic transcriptional programs of immature HPCs, which are programed to be silenced during differentiation. On the other hand, MYC upregulates genes involved with de novo nucleotide/protein synthesis. This is in line with the known involvement of MYC in proliferation and associated anabolic processes (*Ji et al., 2011*). Furthermore, MLL-AF10 activated both the HOXA9 high and MYC high signature genes to induce leukemia while ectopic expression of HOXA9 and MYC synergistically induced leukemia. Thus, our findings suggest that MLL fusion proteins activate both HOXA9- and MYC-dependent programs as alternative mechanisms to the combination of class I and II mutations.

Although HOXA9 maintained MYC expression to immortalize myeloid progenitors ex vivo, it did not induce leukemia in vivo. We speculate that HOXA9 alone cannot maintain MYC expression at a level sufficient to confer leukemogenic ability in vivo. MLL fusion- and HOXA9/MEIS1-transduced cells, which are capable of inducing leukemia in vivo, expressed *Myc* 20–100% more than HOXA9-ICs (*Figure 1C* and *Figure 4—figure supplement 1A*). This additional MYC activity may be required to achieve sufficient leukemogenic ability, therefore MYC/HOXA9-transduced cells were able to induce leukemia in vivo (*Figure 4E*). It should be noted that MYC/HOXA9-ICs and -LCs expressed endogenous *Myc* at a lower level than other HOXA9-expressing cells (i.e. MLL-AF10- and HOXA9/MEIS1-ICs/LCs) (*Figure 4—figure supplement 2*), indicating that there is a threshold of MYC expression a cell can endure. We speculate that HOXA9 uplifts the threshold of MYC expression by conferring anti-apoptotic properties but cannot hyper-activate MYC expression by itself. Thus, additional MYC activation mediated by MLL fusions and MEIS1 confers proliferative advantages sufficient to induce leukemia in vivo.

The therapeutic efficacy of BCL2 inhibitor has been reported in AML including MLL fusion-mediated leukemia (*Pan et al., 2014*). Because there is a correlation between the expression levels of HOX proteins and the sensitivity to BCL2 inhibitor in AML patient samples, the aberrant expression of HOXA9 is the potential mechanism for BCL2-dependence of MLL leukemia (*Brumatti et al., 2013*; *Kontro et al., 2017*). Oncogenic MYC expression often leads to apoptosis, which needs to be alleviated by additional genetic events for leukemic cell survival (*McMahon, 2014*). Indeed, co-expression of HOXA9, BCL2, or SOX4 promoted MYC-mediated leukemogenesis, while MYC alone was insufficient to induce leukemia in vivo. Although the downstream mechanisms could not be addressed, SOX4 also exhibited anti-apoptotic effects on MYC-expressing cells. BCL2 was highly expressed in MLL fusion-mediated leukemia (*Figure 5—figure supplement 2A,B*), and many human MLL fusion leukemia cell lines were sensitive to a BCL-2 inhibitor (i.e. ABT-199)(*Pan et al., 2014*). High expression of SOX4 is correlated to poor survival in AML patients (*Lu et al., 2017*). These results support the key roles of BCL2 and SOX4 in the development of MLL leukemia. Ectopic expression of MYC and BCL2 induced lymphoid leukemia, whereas the MYC/SOX4 combination exclusively induced myeloid leukemia (*Figure 6—figure supplement 1B*), suggesting that BCL2 confers relatively stronger survival advantages in the lymphoid lineage than in the myeloid lineage in vivo, while SOX4 does in the myeloid lineage. Importantly, there was a partial effect of single-gene knockout of *Bcl2* and *Sox4* on leukemia initiation and maintenance. This indicates that MLL fusion proteins exploit multiple anti-apoptotic pathways and that blocking a single anti-apoptotic pathway may be insufficient to completely abrogate leukemic potential. Thus, simultaneously blocking multiple anti-apoptotic pathways may be required for efficient molecularly targeted therapy of HOXA9-expressing leukemia.

Our results also provide an insight into the mode of function of HOXA9. *HOX* genes are known to be expressed in a position-specific manner, conferring a positional identity to a cell (*Wang et al., 2009*). Our results suggest that HOXA9 unlikely functions as a major upstream factor determining tissue-specific gene expression by turning a silenced chromatin into transcriptionally active chromatin. Rather, HOX proteins likely play a supportive role in maintaining gene expression, which was activated by other transcriptional regulators. Our observation that HOXA9 cannot reactivate gene expression once silenced fits this hypothesis. Accordingly, HOXA9 maintains a subset of genes related to hematopoietic identity when expressed in hematopoietic precursors (*Figure 2B*). Recently, it has been reported that HOXA9 may recruit enhancer apparatuses, as it colocalizes with active enhancer mark (i.e. acetylated histone H3 lysine 27) (*Sun et al., 2018*). We speculate that HOXA9 may support the maintenance of an active enhancer, but unlikely establishes it on silenced chromatin. Further functional analysis of HOXA9 is required to understand how HOXA9 regulates gene expression.

In summary, our results describe the oncogenic roles for HOXA9 as a transcriptional maintenance factor for multiple anti-apoptotic genes, which are necessary to promote MYC-mediated leukemogenesis. In the case of MLL fusion-mediated leukemia, MLL fusion proteins directly activate both *MYC* and *HOXA9*, while HOXA9 maintains expression of *MYC*, *BCL2*, and *SOX4*, achieving high MYC activity and anti-apoptotic properties simultaneously (*Figure 7E*). Thus, MLL fusion-mediated LCs possess highly proliferative potentials and survival advantages using HOXA9 as a critical mediator.

## Materials and methods

### Vector constructs

For protein expression vectors, cDNAs obtained from Kazusa Genome Technologies Inc (*Nagase et al., 2008*) were modified by PCR-mediated mutagenesis and cloned into the pMSCV vector (for virus production) or pCMV5 vector (for transient expression) by restriction enzyme digestion and DNA ligation. The MSCV-neo MLL-ENL and MLL-AF10 vectors have been previously described (*Okuda et al., 2017*). sgRNA-expression vectors were constructed using the pLKO5. sgRNA.EFS.GFP vector (RRID:Addgene_57822) (*Heckl et al., 2014*). shRNA-expression vectors were purchased from Dharmacon. The target sequences are listed in *Supplementary file 1*.

## Cell lines

HEK293T cells were a gift from Michael Cleary and were authenticated by the JCRB Cell Bank in 2019 (Key Resources Table). HEK293TN cells were purchased from System Biosciences (RRID:CVCL_UL49). The cells were cultured in Dulbecco's modified Eagle's medium (DMEM) supplemented with 10% fetal bovine serum (FBS) and penicillin-streptomycin (PS). The Platinum-E (PLAT-E) ecotropic virus packaging cell line—a gift from Toshio Kitamura (RRID:CVCL_B488)(Morita et al., 2000)—was cultured in DMEM supplemented with 10% FBS, puromycin, blasticidin, and PS. The human leukemia cell lines including HB1119, (RRID:CVCL_8227), K562 (RRID:CVCL_0004), RS4-11 (RRID:CVCL_0093), THP1 (RRID:CVCL_0006), and EOL1 (RRID:CVCL_0258) were gifts from Michael Cleary (Tkachuk et al., 1992; Yokoyama et al., 2004), and were cultured in RPMI 1640 medium supplemented with 10% FBS and PS. The MV4-11 cell line (ATCC, RRID:CVCL_0064) was cultured in Iscove's modified Dulbecco's medium (IMDM) supplemented with 10% FBS and PS. The ML-2 cell line (DSMZ, RRID:CVCL_1418) was cultured in RPMI 1640 medium supplemented with 10% FBS and PS. MLL-ENL LCs (MLL-ENLbm0713) were described previously (Okuda et al., 2017). All the cell lines except HEK293TN, PLAT-E, and HB1119 were authenticated using short tandem repeat (STR)-PCR method by the JCRB Cell Bank. Cells were cultured in the incubator at 37°C and 5% $CO_2$, and routinely tested for mycoplasma using the MycoAlert Mycoplasma detection kit (Lonza).

## Animal models

For the in vivo leukemogenesis assay, 8-week-old female C57BL/6JJcl (C57BL/6J) or C. B-17/Icr-*scid/scid*Jcl (SCID) mice were purchased from CLEA Japan (Tokyo, Japan). *Bcl2*-knockout mice were a gift from Yoshihide Tsujimoto and provided via RIKEN BRC (Kamada et al., 1995). *Sox4*-knockout mice were a gift from Hans Clevers and provided via RIKEN BRC (Schilham et al., 1996).

## Western blotting

Western blotting was performed as previously described (Yokoyama et al., 2002). Antibodies used in this study are listed in Key Resources Table.

## Virus production

Ecotropic retrovirus constructed in pMSCV vectors was produced using PLAT-E packaging cells (Morita et al., 2000). Lentiviruses were produced in HEK293TN cells using the pMDLg/pRRE (RRID:Addgene_12251), pRSV-rev (RRID:Addgene_12253), and pMD2.G (RRID:Addgene_12259) vectors, all of which were gifts from Didier Trono (Dull et al., 1998). The virus-containing medium was harvested 24–48 hr following transfection and used for viral transduction.

## Myeloid progenitor transformation assay

The myeloid progenitor transformation assay was conducted as previously described (Lavau et al., 1997; Okuda and Yokoyama, 2017a). Bone marrow cells were harvested from the femurs and tibiae of 5-week-old female C57BL/6J mice. c-Kit$^+$ cells were enriched using magnetic beads conjugated with an anti-c-Kit antibody (Miltenyi Biotec, RRID:AB_2753213), transduced with a recombinant retrovirus by spinoculation, and then plated (4 × 10$^4$ cells/ sample) in a methylcellulose medium (IMDM, 20% FBS, 1.6% methylcellulose, and 100 μM β-mercaptoethanol) containing murine stem cell factor (mSCF), interleukin 3 (mIL-3), and granulocyte-macrophage colony-stimulating factor (mGM-CSF; 10 ng/mL each). During the first culture passage, G418 (1 mg/mL) or puromycin (1 μg/mL) was added to the culture medium to select for transduced cells. *Hoxa9* expression was quantified by qRT-PCR after the first passage. Cells were then re-plated once every 4–6 days with fresh medium; the number of plated cells for the second, third, and fourth passages was 4 × 10$^4$, 2 × 10$^4$, and 1 × 10$^4$ cells/well, respectively. CFUs were quantified per 10$^4$ plated cells at each passage.

## In vivo leukemogenesis assay

In vivo leukemogenesis assays were conducted as previously described (Lavau et al., 1997; Okuda and Yokoyama, 2017b). c-Kit$^+$ cells (2 × 10$^5$) prepared from the femurs and tibiae of 5-week-old female C57BL/6J mouse were transduced with retrovirus by spinoculation and intravenously transplanted into sublethally irradiated (5–6 Gy) C57BL/6J mice. For secondary leukemia, LCs (2 × 10$^5$) cultured ex vivo for more than three passages were transplanted. As for knockouts of *Bcl2*

and *Sox4*, mice heterozygous for *Bcl2* or *Sox4* were crossed, and c-Kit$^+$ cells were isolated from fetal livers at E14–15 (for *Bcl2*) or E13 (for *Sox4*). The next day, cells were transduced with the retroviruses for MLL-AF10 or HOXA9/MEIS1 and transplanted intravenously into sublethally irradiated (2.5 Gy) SCID mice [$2 \times 10^5$ (for *Bcl2*) or $1 \times 10^5$ cells/mouse (for *Sox4*)].

## qRT-PCR

Total RNA was isolated using the RNeasy Mini Kit (Qiagen) and reverse-transcribed using the Super-script III First Strand cDNA Synthesis System (Thermo Fisher Scientific) with oligo (dT) primers. Gene expression was analyzed by qPCR using TaqMan probes (Thermo Fisher Scientific). Relative expression levels were normalized to those of *GAPDH/Gapdh, ACTB/Actb,* or *TBP/Tbp* and determined using a standard curve and the relative quantification method, according to manufacturer's instructions (Thermo Fisher Scientific). Commercially available/custom made PCR probes used are listed in *Supplementary file 1*.

## ChIP-qPCR and ChIP-seq

ChIP was performed as previously described (*Okuda et al., 2017*), using the fanChIP method (*Miyamoto and Yokoyama, 2021*). DNA was precipitated with glycogen, dissolved in TE buffer, and analyzed by qPCR (ChIP-qPCR) or deep sequencing (ChIP-seq). The qPCR probe/primer sequences are listed in *Supplementary file 1*. Deep sequencing was performed using the TruSeq ChIP Sample Prep Kit (Illumina) and HiSeq2500 (Illumina) at the core facility of Hiroshima University and described in our previous publication (*Okuda et al., 2017*).

## RNA-seq

Total RNA was prepared using the RNeasy Kit (Qiagen) and analyzed using a Bioanalyzer (Agilent Technologies). Deep sequencing was performed using a SureSelect Strand Specific RNA Library Prep Kit (Agilent Technologies) and GAIIx (Illumina) with 36 bp single-end reads or HiSeq2500 (Illumina) with 51 bp single-end reads at the core facility of Hiroshima University. Sequenced reads were mapped to the human genome assembly hg19 or the mouse genome assembly mm9 using CASAVA 1.8.2 (Illumina, RRID:SCR_001802), and read counts were normalized as reads per kilo base of exon per million mapped (RPKM). To define HOXA9 high and MYC high signature genes, data were trimmed by removing lowly expressed genes whose RPKM values were less than 2, and the relative expression between HOXA9-ICs and MYC-ICs was estimated. The top 50 HOXA9 high and MYC high signature genes were visualized as a heatmap using the Complex Heatmap package (RRID: SCR_017270)(*Gu et al., 2016*). To define HOXA9-MYC common target genes, lowly expressed genes were removed, and genes with more than two-fold expression compared to c-Kit$^+$ cells were determined for each of HOXA9-ICs and MYC-ICs. The overlapping genes were defined as 'HOXA9-MYC common target genes'. GSEA analysis and Kyoto Encyclopedia of Genes and Genomes (KEGG) pathway analysis were performed using the DAVID (RRID:SCR_001881)(*Jiao et al., 2012*), GSEA (*Subramanian et al., 2005*), and Metascape (*Zhou et al., 2019*).

## sgRNA competition assay

*Cas9* was introduced via lentiviral transduction using the pKLV2-EF1a-Cas9Bsd-W vector (RRID: Addgene_68343) (*Tzelepis et al., 2016*). Cas9-expressing stable lines were established with blastici-din (10–30 µg/mL) selection. The targeting sgRNA was co-expressed with GFP via lentiviral transduction using pLKO5.sgRNA.EFS.GFP vector (RRID:Addgene_57822)(*Heckl et al., 2014*). Percentages of GFP$^+$ cells were initially determined by FACS analysis at 2 or 3 days after sgRNA transduction and then measured once every 3–5 days.

## FACS analysis and sorting

To detect apoptosis, two to five million cells were suspended in 200 µL of reaction buffer (140 mM NaCl, 10 mM HEPES, 2.5 mM CaCl$_2$, and 0.1% BSA) and incubated with APC-Annexin V (RRID:AB_2868885) for 15 min and Propidium iodide for 5 min at room temperature. The cells were then centrifuged, resuspended in fresh reaction buffer, and analyzed with FACS Melody (BD Bioscience). FACS sorting of mouse bone marrow cells was performed with fluorophore-conjugated antibodies listed in Key Resources Table as previously described (*Yokoyama et al., 2013*).

## Accession numbers

Deep sequencing data used in this study have been deposited in the DNA Data Bank of Japan (DDBJ) Sequence Read Archive under the accession numbers listed in *Supplementary file 1*.

## Statistics

Statistical analyses were performed using GraphPad Prism seven software (RRID:SCR_002798). Data are presented as the mean with standard deviation (SD). Comparisons between two groups were analyzed by unpaired two-tailed Student's *t*-test, while multiple comparisons were performed by ordinary one-way analysis of variance (ANOVA) followed by Dunnett's test or two-way ANOVA. Mice transplantation experiments were analyzed by the log-rank test and Bonferroni correction was applied for multiple comparisons. p Values < 0.05 were considered statistically significant. n.s.: $p > 0.05$, *: $p \leq 0.05$, **: $p \leq 0.01$, ***: $p \leq 0.001$, and ****: $p \leq 0.0001$.

## Study approval

All animal experimental protocols were approved by the National Cancer Center (Tokyo Japan) Institutional Animal Care and Use Committee.

## Acknowledgements

We thank Yuzo Sato, Boban Stanojevic, Makiko Okuda, Megumi Nakamura, Etsuko Kanai, Aya Nakayama, Hagumu Sato, Ikuko Yokoyama, Kanae Ito, and Ayako Yokoyama for technical assistance. We thank Drs. Yoshihide Tsujimoto and Hans Clevers for providing us the knockout mouse lines of *Bcl2* and *Sox4*, respectively. These mouse lines were provided by the RIKEN BRC through the National BioResource Project of the MEXT, Japan. We also thank all members of the Shonai Regional Industry Promotion Center for their administrative support. This work was supported by the Japan Society for the Promotion of Science (JSPS) KAKENHI grants (16H05337 and 19H03694 to AY; 19K16791 to RM) and in part by research funds from the Yamagata prefectural government, the City of Tsuruoka, Dainippon Sumitomo Pharma Co. Ltd., and the Friends of Leukemia Research Fund.

## Additional information

### Competing interests

Akihiko Yokoyama: received a research grant from Dainippon Sumitomo Pharma Co. Ltd. The other authors declare that no competing interests exist.

### Funding

| Funder | Grant reference number | Author |
| --- | --- | --- |
| Japan Society for the Promotion of Science | 16H05337 | Akihiko Yokoyama |
| Japan Society for the Promotion of Science | 19H03694 | Akihiko Yokoyama |
| Japan Society for the Promotion of Science | 19K16791 | Ryo Miyamoto |
| The Yamagata Prefectural Government | Research grant | Akihiko Yokoyama |
| The City of Tsuruoka | Research grant | Akihiko Yokoyama |
| Dainippon Sumitomo Pharma Co., Ltd. | Research grant | Akihiko Yokoyama |
| The Friends of Leukemia Research Fund | Research grant | Akihiko Yokoyama |

The funders had no role in study design, data collection and interpretation, or the decision to submit the work for publication.

## Author contributions
Ryo Miyamoto, Data curation, Formal analysis, Funding acquisition, Validation, Investigation, Visualization, Methodology, Writing - original draft, Writing - review and editing; Akinori Kanai, Yosuke Komata, Data curation, Formal analysis, Investigation, Visualization; Hiroshi Okuda, Data curation, Formal analysis, Investigation; Satoshi Takahashi, Formal analysis, Investigation; Hirotaka Matsui, Toshiya Inaba, Resources, Formal analysis, Investigation, Methodology; Akihiko Yokoyama, Conceptualization, Resources, Data curation, Formal analysis, Supervision, Funding acquisition, Validation, Investigation, Visualization, Methodology, Writing - original draft, Project administration, Writing - review and editing

## Author ORCIDs
Akinori Kanai (ID) https://orcid.org/0000-0003-1555-6768
Akihiko Yokoyama (ID) https://orcid.org/0000-0002-5639-8068

## Ethics
Animal experimentation: All animal experimental protocols were approved by the National Cancer Center (Tokyo Japan) Institutional Animal Care and Use Committee.

## Decision letter and Author response
Decision letter https://doi.org/10.7554/eLife.64148.sa1
Author response https://doi.org/10.7554/eLife.64148.sa2

---

# Additional files

## Supplementary files
- Source data 1. Gel images.
- Supplementary file 1. Reagents and data sets.
- Transparent reporting form

## Data availability
ChIP-seq data have been deposited to the DDBJ archive and have been published (accession number: DRA004871). RNA-seq data have been deposited to the DDBJ archive and have been published (accession number: DRA010090, DRA012078, DRA004874, DRA012079).

The following datasets were generated:

| Author(s) | Year | Dataset title | Dataset URL | Database and Identifier |
|---|---|---|---|---|
| HIROSHIMA | 2020 | Expression profiles of murine myeloid progenitors immortalized by various oncogenes | https://ddbj.nig.ac.jp/public/ddbj_database/gea/experiment/E-GEAD-000/E-GEAD-360/ | DDBJ GEA, E-GEAD-360 |
| HIROSHIMA | 2021 | Expression profiles of murine myeloid progenitors immortalized by various oncogenes | https://ddbj.nig.ac.jp/public/ddbj_database/gea/experiment/E-GEAD-000/E-GEAD-435/ | DDBJ GEA, E-GEAD-435 |
| HIROSHIMA | 2021 | Expression profiles of human cell lines | https://ddbj.nig.ac.jp/public/ddbj_database/gea/experiment/E-GEAD-000/E-GEAD-436/ | DDBJ GEA, E-GEAD-436 |
| HIROSHIMA | 2018 | Sequence reads of murine myeloid progenitors immortalized by various oncogenes | https://ddbj.nig.ac.jp/DRASearch/submission?acc=DRA010090 | DDBJ DRA, DRA010090 |
| HIROSHIMA | 2015 | Sequence reads of murine myeloid progenitors immortalized by various oncogenes | https://ddbj.nig.ac.jp/DRASearch/submission?acc=DRA012078 | DDBJ DRA, DRA012078 |

| | | | | |
|---|---|---|---|---|
| HIROSHIMA | 2016 | Sequence reads of various factors/modificaions in HB1119 cells | https://ddbj.nig.ac.jp/DRASearch/submission?acc=DRA004874 | DDBJ DRA, DRA00 4874 |
| HIROSHIMA | 2015 | Sequence reads of human cell lines | https://ddbj.nig.ac.jp/DRASearch/submission?acc=DRA012079 | DDBJ DRA, DRA0120 79 |
| HIROSHIMA | 2020 | Expression profiles of HB1119 and 293T cell lines | https://ddbj.nig.ac.jp/public/ddbj_database/gea/experiment/E-GEAD-000/E-GEAD-321/ | DDBJ GEA, E-GEAD-321 |

The following previously published datasets were used:

| Author(s) | Year | Dataset title | Dataset URL | Database and Identifier |
|---|---|---|---|---|
| HIROSHIMA | 2016 | Genomic localization of various factors/modificaions in HB1119 cells | https://ddbj.nig.ac.jp/DRASearch/submission?acc=DRA004871 | DDBJ, DRA004871 |
| HIROSHIMA | 2020 | Genomic localization of various factors/modificaions in HB1119 cells | https://ddbj.nig.ac.jp/public/ddbj_database/gea/experiment/E-GEAD-000/E-GEAD-319/ | DDBJ GEA, E-GEAD-319 |

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

# Appendix 1

**Appendix 1—key resources table**

| Reagent type (species) or resource | Designation | Source or reference | Identifiers | Additional information |
|---|---|---|---|---|
| Strain, strain background (*Mus musculus*) | C57BL/6J | CLEA Japan | | |
| Strain, strain background (*Mus musculus*) | Bcl2 KO (B6.129P2-Bcl2 <tm1Tsu>/ TsuRbrc) | Gift from Yoshihide Tsujimoto (via RIKEN BRC) *Kamada et al., 1995* | | |
| Strain, strain background (*Mus musculus*) | Sox4 KO (STOCK Sox4 <tm1Cle>/ Mmmh) | Gift from Hans Clevers (via RIKEN BRC) *Schilham et al., 1996* | | |
| Cell line (Homo-sapiens) | PLAT-E | Gift from Toshio Kitamura *Morita et al., 2000* | RRID:CVCL_B488 | |
| Cell line (Homo-sapiens) | HB1119 | Gift from Michael L. Cleary *Tkachuk et al., 1992*; *Yokoyama et al., 2004* | | The cell line was verified by the expression of MLL-ENL |
| Cell line (Homo-sapiens) | HEK293TN | System Bioscience | Cat# LV900A-1 RRID:CVCL_UL49 | |
| Cell line (Homo-sapiens) | HEK293T | Gift from Michael L. Cleary *Yokoyama et al., 2004* | | authenticated by the JCRB Cell Bank in 2019 |
| Cell line (Homo-sapiens) | K562 | Gift from Michael L. Cleary *Yokoyama et al., 2004* | RRID:CVCL_0004 | authenticated by the JCRB Cell Bank in 2021 |
| Cell line (Homo-sapiens) | MV4-11 | ATCC | Cat# CRL-9591 RRID:CVCL_0064 | authenticated by the JCRB Cell Bank in 2021 |
| Cell line (Homo-sapiens) | RS4-11 | Gift from Michael L. Cleary | RRID:CVCL_0093 | authenticated by the JCRB Cell Bank in 2021 |
| Cell line (Homo-sapiens) | ML-2 | DSMZ | Cat# ACC15 RRID:CVCL_1418 | authenticated by the JCRB Cell Bank in 2021 |
| Cell line (Homo-sapiens) | THP1 | Gift from Michael L. Cleary | RRID:CVCL_0006 | authenticated by the JCRB Cell Bank in 2021 |
| Cell line (Homo-sapiens) | EOL-1 | Gift from Michael L. Cleary | RRID:CVCL_0258 | authenticated by the JCRB Cell Bank in 2021 |
| Antibody | MLLn ab#1 (Rabbit polyclonal) | In-house *Yokoyama et al., 2002* | rpN1 | ChIP (1:400) |

*Continued on next page*

*Appendix 1—key resources table continued*

| Reagent type (species) or resource | Designation | Source or reference | Identifiers | Additional information |
|---|---|---|---|---|
| Antibody | MLLn ab#2 (Rabbit monoclonal, D2M7U) | Cell Signaling Technology | Cat# 14689 RRID:AB_2688009 | ChIP (1:400) |
| Antibody | MLLc (Rabbit monoclonal, D6G8N) | Cell Signaling Technology | Cat# 14197 RRID:AB_2688010 | WB (1:1000) |
| Antibody | MENIN (Rabbit polyclonal) | Bethyl Laboratories | Cat# A300-105A RRID:AB_2143306 | ChIP (1:400) WB (1:1000) |
| Antibody | MYC (Rabbit monoclonal, D3N8F) | Cell Signaling Technology | Cat# 13987 RRID:AB_2631168 | WB (1:2000) |
| Antibody | Caspase3 (Rabbit monoclonal, D3R6Y) | Cell Signaling Technology | Cat# 14220 RRID:AB_2798429 | WB (1:2000) |
| Antibody | Cleaved Caspase3 (Rabbit monoclonal, 5A1E) | Cell Signaling Technology | Cat# 9664 RRID:AB_2070042 | WB (1:2000) |
| Antibody | PARP (Rabbit polyclonal) | Cell Signaling Technology | Cat# 9542 RRID:AB_2160739 | WB (1:2000) |
| Antibody | γH2AX (Rabbit polyclonal) | Bethyl Laboratories | Cat# A300-081A RRID:AB_203288 | WB (1:1000) |
| Antibody | BCL2 (Mouse monoclonal, C-2) | Santa Cruz Biotechnology | Cat# sc-7382 RRID:AB_626736 | WB (1:500) |
| Antibody | SOX4 (Mouse monoclonal, B-7) | Santa Cruz Biotechnology | Cat# sc-518016 | WB (1:200) |
| Antibody | HOXA9 (Rabbit polyclonal) | Millipore | Cat# 07–178 RRID:AB_11210179 | WB (1:1000) |
| Antibody | GAPDH (Rabbit polyclonal) | Santa Cruz Biotechnology | Cat# sc-25778 RRID:AB_10167668 | WB (1:2000) |
| Antibody | BAK (Rabbit monoclonal, D4E4) | Cell Signaling Technology | Cat# 12105 RRID:AB_2716685 | WB (1:2000) |
| Antibody | BAD (Rabbit monoclonal, D24A9) | Cell Signaling Technology | Cat# 9239 RRID:AB_2062127 | WB (1:2000) |
| Antibody | BCL-XL (Rabbit monoclonal, 54H6) | Cell Signaling Technology | Cat# 2764 RRID:AB_2228008 | WB (1:2000) |

*Continued on next page*

*Appendix 1—key resources table continued*

| Reagent type (species) or resource | Designation | Source or reference | Identifiers | Additional information |
|---|---|---|---|---|
| Antibody | MCL1 (Rabbit monoclonal, D35A5) | Cell Signaling Technology | Cat# 5453 RRID:AB_10694494 | WB (1:2000) |
| Antibody | APC Annexin V | BD Biosciences | Cat# 550475 RRID:AB_2868885 | FACS (1:100) |
| Antibody | Propidium iodide | Thermo Fisher Scientific | P3566 | FACS (1:2000) |
| Antibody | Gr-1 (Rat monoclonal, RB6-8C5) | BD Biosciences | Cat# 553127 RRID:AB_394643 | FACS (1:100) |
| Antibody | B220 (Rat monoclonal, RA3-6B2) | BD Biosciences | Cat#: 553088 RRID:AB_394618 | FACS (1:100) |
| Antibody | TER119 (Rat monoclonal, TER-119) | BD Biosciences | Cat#: 557915 RRID:AB_396936 | FACS (1:100) |
| Antibody | CD3e (Hamster monoclonal, 145–2 C11) | BD Biosciences | Cat#: 553062 RRID:AB_394595 | FACS (1:100) |
| Antibody | Mac/CD11b (Rat monoclonal, M1/70) | BD Biosciences | Cat#: 553310 RRID:AB_394774 | FACS (1:100) |
| Antibody | c-Kit (Rat monoclonal, 2B8) | BD Biosciences | Cat#: 553355 RRID:AB_394806 | FACS (1:100) |
| Antibody | Sca1 (Rat monoclonal, D7) | BD Biosciences | Cat#: 553108 RRID:AB_394629 | FACS (1:100) |
| Antibody | CD34 (Rat monoclonal, RAM34) | BD Biosciences | Cat#: 751621 RRID:AB_2875614 | FACS (1:100) |
| Antibody | CD16/32 (Rat monoclonal, 93) | BD Biosciences | Cat#: 751690 RRID:AB_2875675 | FACS (1:100) |
| Recombinant DNA reagent | pMSCV-neo | Clontech | Cat#: 634401 | Gene expression vector |
| Recombinant DNA reagent | pMSCV-puro | Clontech | Cat#: 634401 | Gene expression vector |
| Recombinant DNA reagent | pMSCV-hygro | Clontech | Cat#: 634401 | Gene expression vector |
| Recombinant DNA reagent | pLKO5.EFS.GFP | Addgene (gift from Benjamin Ebert) *Heckl et al., 2014* | Addgene Plasmid #57822 RRID:Addgene_57822 | sgRNA backbone |
| Recombinant DNA reagent | pKLV2-Cas9.bsd | Addgene (gift from Kosuke Yusa) *Tzelepis et al., 2016* | Addgene Plasmid #68343 RRID:Addgene_68343 | Cas9 expression vector |
| Recombinant DNA reagent | pMDLg/pRRE | Addgene (gift from Didier Trono) *Dull et al., 1998* | Addgene Plasmid #12251 RRID:Addgene_12251 | Lentivirus packaging vector |

*Appendix 1—key resources table continued*

| Reagent type (species) or resource | Designation | Source or reference | Identifiers | Additional information |
|---|---|---|---|---|
| Recombinant DNA reagent | pRSV-rev | Addgene (gift from Didier Trono) *Dull et al., 1998* | Addgene Plasmid #12253 RRID:Addgene_12253 | Lentivirus packaging vector |
| Recombinant DNA reagent | pMD2.G | Addgene (gift from Didier Trono) *Dull et al., 1998* | Addgene Plasmid #12259 RRID:Addgene_12259 | Lentivirus packaging vector |
| Recombinant DNA reagent | pLKO.1-puro | Addgene (gift from Bob Weinberg) *Stewart et al., 2003* | Addgene Plasmid #84530 RRID:Addgene_84530 | shRNA backbone |
| Sequence-based reagent | sgRNAs | This study | | See *Supplementary file 1* |
| Sequence-based reagent | shRNA | This study | | See *Supplementary file 1* |
| Sequence-based reagent | qPCR primers | Thermo Fisher Scientific | | See *Supplementary file 1* |
| Commercial assay or kit | RNeasy Mini Kit | Qiagen | Cat#: 74106 | |
| Commercial assay or kit | Super Script III First-Strand Synthesis System | Thermo Fisher Scientific | Cat# 18080051 | |
| Commercial assay or kit | TruSeq ChIP Sample Prep Kit SetB | Illumina | Cat# IP-202–1024 | |
| Commercial assay or kit | Sure Select Strand Specific RNA Library Prep Kit | Agilent Technologies | Cat#: G9691A | |
| Software, algorithm | GraphPad Prism7 | GraphPad Software Inc | RRID:SCR_002798 | Data analysis |
| Software, algorithm | FlowJo | BD Biosciences | RRID:SCR_008520 | FACS data analysis |
| Software, algorithm | Integrative Genomics Viewer | | *Thorvaldsdóttir et al., 2013* | RRID:SCR_011793 |
| Data visualization | | | | |
| Software, algorithm | CASAVA 1.8.2 | Illumina | RRID:SCR_001802 | RNA-seq data analysis |
| Software, algorithm | DAVID | *Jiao et al., 2012* | RRID:SCR_001881 | RNA-seq data analysis |
| Software, algorithm | Complex heatmap | *Gu et al., 2016* | RRID:SCR_017270 | RNA-seq data analysis |
| Software, algorithm | R2: Genome Analysis and Visualization Platform | AMC: Oncogenomics | http://r2.amc.nl http://r2platform.com | RNA-seq data analysis |
| Software, algorithm | Gene set enrichment analysis | *Subramanian et al., 2005* | https://www.gsea-msigdb.org/gsea/index.jsp | RNA-seq data analysis |
| Other | c-Kit magnetic beads | Miltenyi Biotec | Cat# 130-091-224 RRID:AB_2753213 | |

