## [Decision Letter]

**Acceptance summary:**

The manuscript of Miyamoto et al. describes the synergistic interactions between HOXA9 and MYC induced by MLL-AF10 fusions in myeloid leukemogenesis. Detailed analysis of gene expression profiles resulting from overexpression of MLL-AF10 provide mechanistic insight into how cell death induced by elevated MYC is counted by signaling from BCL2 or SOX4, which are up-regulated in HOXA9, supporting leukemogenesis. This manuscript will be of interest to experimental haematologists studying initiation and maintenance factors in leukaemia.

**Decision letter after peer review:**

Thank you for submitting your article "HOXA9 promotes MYC-mediated leukemogenesis by maintaining gene expression for multiple anti-apoptotic pathways" for consideration by *eLife*. Your article has been reviewed by 2 peer reviewers, and the evaluation has been overseen by Erica Golemis as the Senior and Reviewing Editor. The reviewers have opted to remain anonymous.

Essential revisions:

1. Presentation of ChIPseq data in Figure 1 could be substantially improved by providing more details of MLL-occupancy patterns. More information (particularly about computational analyses) should also be provided in the methods For example, in Figure 1A, the authors attempt to identify direct target genes of the MLL fusion protein MLL-ENL by performing ChIPseq using an anti-MLL antibody. Whether or not the signal can be attributed to MLL-ENL or wild-type MLL is unclear. Furthermore, genome-wide MLL-occupancy patterns are not shown. It would also be useful to reconcile current data with other publicly available datasets for MLL or MLL-fusion protein occupancy in comparable contexts.

2. It would appear (based on capitalisation), that the authors are over-expressing human transgenes in mouse cells. This is not necessarily a concern, but should be considered when interpreting the data. Likewise, whether the primers used for qPCR are detecting expression of the transgenes, the endogenous genes or both is important (for some of the figures such as Figure 1C there seems to be a mix e.g. Myc vs HoxA9/HOXA9)?

3. Most of the in vivo transplantation experiments have not been performed using fluorescent reporters or congenic recipients that would enable identification of donor-derived cells. Differences between the groups could be attributed to differential engraftment, or potentially even immune rejection (assuming ectopic expression of human transgenes in an immune-competent context). Disease features in recipient mice (beyond survival) are also not shown and expression of transgenes at end-point not confirmed.

4. The authors claim that the data in Figure 5B confirms direct regulation of Bcl2, Sox4 and Igf1 by HOXA9. However, the regulation could also be indirect e.g. HOXA9 could regulate a transcription factor that regulates those genes, or HOXA9 depletion could induce differentiation that may result in downregulation of those genes.

5. More specifically, the authors showed that HOXA9 introduction into MYC-IC failed to show activation of HOXA9 target genes. Although the authors claim that HOXA9 is a transcription maintenance factor with these results (page 14), they did not mention the possible difference of cell-of-origin. A number of oncogenes that function as chromatin remodeling factors failed to show such functions if they are introduced inappropriate target cells. Therefore, the title "HOXA9 functions as a transcription maintenance factor" should be modified. Also, they argue that HOXA9 is not a major upstream factor since the similarly functional fibroblasts express different HOX genes. This notion is too speculative and should be modified or removed.

6. The role of MYC, HOXA9 and BCL2 has been extensively studied in AML including with sophisticated in vivo models utilising conditional alleles. Likewise, many studies have sought to identify essential genes downstream of MLL fusion proteins. A lot of additional data is publicly available (e.g. RNAseq in AML patients) the analysis of which could be used in many different ways to strengthen the manuscript.

Inclusion of analysis based on these data should be considered.

7. The authors showed that Myc expression is comparable between HOXA9-IC and MLL-ENL-IC (Figure 1C), indicating that both HOXA9 and MLL-ENL could support Myc expression as is also argued by authors. This information is inconsistent with the results that co-expression of HOXA9 and MYC induces leukemia whereas HOXA9 single expression does not (Figure 4E). The authors should explain this apparent discrepancy.

8. Moreover, Myc silencing at ~50% expression level of control completely abrogated colony formation in MLL-ENL-IC and HOXA9-IC (Figure 1E). Is there fine-tuning system that requires critical expression level of Myc? Or did the authors obtain much more silencing effect at the protein level? Please clarify this point.

9. In relation to the above questions, given that Myc expression is upregulated by HOXA9 like by MLL-AF10, there should be common genetic pathways regulated both HOXA9 and MYC. I would request the authors to provide the gene list and pathways commonly regulated by HOXA9 and MYC in Figure 2.

10. In discussion (page 21, lines 6-8) the authors claim that MLL fusion proteins promote oncogenesis by activating both MYC- and HOXA9-related pathways. However, the cooperative effect of MYC for HOXA9 is much weaker than that of MEIS1 as is shown in Figure 4E. Given that MEIS1 is the direct target of MLL fusions, I wonder whether MYC is dispensable in the presence of MEIS1. Since there have been several studies on MEIS1 function in leukemogenesis, the authors should show common and/or distinct downstream genes/pathways between MYC- and MEIS1-driven leukemogenesis.

11. BCL2 and SOX4 knockout showed delay in MLL-AF10- and HOXA9/MEIS1-induced leukemogenesis (Figure 7A, B). Please provide the data wheter these delays are caused by increased apoptosis by presenting the Annexin V staining as well as Caspase 3 cleavage and H2AX expression.

---

## [Author Response]

Essential revisions:1. Presentation of ChIPseq data in Figure 1 could be substantially improved by providing more details of MLL-occupancy patterns. More information (particularly about computational analyses) should also be provided in the methods For example, in Figure 1A, the authors attempt to identify direct target genes of the MLL fusion protein MLL-ENL by performing ChIPseq using an anti-MLL antibody. Whether or not the signal can be attributed to MLL-ENL or wild-type MLL is unclear. Furthermore, genome-wide MLL-occupancy patterns are not shown. It would also be useful to reconcile current data with other publicly available datasets for MLL or MLL-fusion protein occupancy in comparable contexts.

We performed ChIP-seq analysis of HB1119 cells in which wildtype MLL, but not MLL-ENL, was specifically knocked down by shRNA (Figure 1A, Figure 1—figure supplement-1B,C), as shown In our previous publication (Okuda et al., 2017). Depletion of wildtype MLL did not affect the ChIP signals. Thus, we concluded that most of the MLL ChIP signals can be attributed to MLL-ENL. These data was presented in our previous report (Okuda et al., 2017) and partially adopted in the revised manuscript. MLL and MLL fusion proteins localize near transcription start sites (TSSs)( Figure 1—figure supplement-1C) because MLL has a CXXC domain that recognizes unmethylated CpGs (Okuda et al., 2014). Such TSS-centric localization of MLL is observed in many other non-MLL-rearranged cell lines such as HEK293T (embryonic kidney) and REH (Leukemia) cells (Miyamoto et al., 2020), in addition to HB1119 cells (MLL-rearranged leukemia cells)(Okuda et al., 2017). We mentioned this in the revised manuscript.

2. It would appear (based on capitalisation), that the authors are over-expressing human transgenes in mouse cells. This is not necessarily a concern, but should be considered when interpreting the data. Likewise, whether the primers used for qPCR are detecting expression of the transgenes, the endogenous genes or both is important (for some of the figures such as Figure 1C there seems to be a mix e.g. Myc vs HoxA9/HOXA9)?

We used human transgenes in the presented experiments. The qPCR probes for mouse *Hoxa9* and *Meis1* detected human *HOXA9* and *MEIS1*, respectively. Hence, we described HOXA9/Hoxa9 and MEIS1/Meis1 to clearly indicate that these probes detect both human and mouse genes. The qPCR probe for mouse endogenous *Myc* did not detect the human *MYC* transgene. The samples producing qPCR signals for both endogenous murine genes and exogenous human transgenes are highlighted by # and faded color (Figure 1C).

3. Most of the in vivo transplantation experiments have not been performed using fluorescent reporters or congenic recipients that would enable identification of donor-derived cells. Differences between the groups could be attributed to differential engraftment, or potentially even immune rejection (assuming ectopic expression of human transgenes in an immune-competent context). Disease features in recipient mice (beyond survival) are also not shown and expression of transgenes at end-point not confirmed.

As for the possibility of immune rejection of the cells expressing human transgenes:

As shown in Figure 3D, the mouse *Myc* gene was tested in addition to human *MYC* and did not induce leukemia in vivo, supporting that the enhanced MYC function alone is insufficient to induce leukemia under these experimental conditions. It has been shown that the mouse *Hoxa9* gene is also a weak oncogene in vivo by Kroon et al. whereas it induced leukemia as a combination with *Meis1*(Kroon et al., 1998). The human HOXA9 transgene phenocopied mouse Hoxa9 in our assays. These results did not support the possibility of immune rejection of the human transgene-expressing cells. We mentioned that in the revised manuscript.

As for the possibility of different engraftment:

We did not mean to exclude the possibility of different engraftment as the reason of not inducing leukemia by a certain oncogene. It is likely that HOXA9 promotes engraftment of MYC-transduced cells by conferring survival advantage with BCL2/SOX4-mediated anti-apoptotic properties. It is possible that HOXA9 mediates additional functions to promote engraftment other than providing anti-apoptotic properties. However, we chose to focus on the HOXA9-mediated anti-apoptotic functions in this paper.

As for the disease features:

We have added the expression and immune phenotype data in Figure 4—figure supplement-3B and Figure 6—figure supplement-1B.

In contrast to MLL-AF10 and HOXA9 containing gene sets (HOXA9-MEIS1, HOXA9-MYC), MYC-BCL2 induced lymphoid leukemia in vivo, consistent with the previous report (Luo et al., 2005). We speculate that HOXA9 and SOX4 are more functional in the myeloid lineage, while BCL2 functions more efficiently in the lymphoid lineage than in the myeloid lineage. Consequently, the MYC-BCL2 combination tended to induce lymphoid leukemia.

As for the expression of the transgene at end-point:

Regarding the expression of the transgenes in Figure 3D and 4E, we have provided the RT-qPCR data for the transgenes in Figure 4—figure supplement-3B.

Regarding the expression of the transgenes in Figure 6B, the protein expression of the transgenes is shown in Figure 6—figure supplement-1A.

Regarding the expression of the transgenes in Figure 7A, B, we have provided the RT-qPCR data for the transgenes in Figure 7—figure supplement 2A.

4. The authors claim that the data in Figure 5B confirms direct regulation of Bcl2, Sox4 and Igf1 by HOXA9. However, the regulation could also be indirect e.g. HOXA9 could regulate a transcription factor that regulates those genes, or HOXA9 depletion could induce differentiation that may result in downregulation of those genes.

The regulatory mechanisms by which HOXA9 controls the expression of its target genes are of great interest. Indeed, the expression of BCL2 and/or SOX4 could be regulated indirectly by HOXA9. We changed the wording by removing the word “direct” in the revised manuscript.

5. More specifically, the authors showed that HOXA9 introduction into MYC-IC failed to show activation of HOXA9 target genes. Although the authors claim that HOXA9 is a transcription maintenance factor with these results (page 14), they did not mention the possible difference of cell-of-origin. A number of oncogenes that function as chromatin remodeling factors failed to show such functions if they are introduced inappropriate target cells. Therefore, the title "HOXA9 functions as a transcription maintenance factor" should be modified. Also, they argue that HOXA9 is not a major upstream factor since the similarly functional fibroblasts express different HOX genes. This notion is too speculative and should be modified or removed.

To test whether the inability to activate HOXA9 target genes by reactivating HOXA9, we performed qRT-PCR analysis of the cells wherein HOXA9 is inactivated and later reactivated within the same cell population (Figure 5—figure supplement-1C). Reactivation of HOXA9 by adding back 4OHT did not rescued the expression of HOXA9 target genes. Thus, we think that the reason why HOXA9 did not activate the expression of its target genes in MYC-transformed cells is not due to the difference of cell-of-origin, rather because HOXA9 is a transcription maintenance factor which cannot initiate the expression from the silenced locus.

6. The role of MYC, HOXA9 and BCL2 has been extensively studied in AML including with sophisticated in vivo models utilising conditional alleles. Likewise, many studies have sought to identify essential genes downstream of MLL fusion proteins. A lot of additional data is publicly available (e.g. RNAseq in AML patients) the analysis of which could be used in many different ways to strengthen the manuscript.Inclusion of analysis based on these data should be considered.

As advised by the reviewer, we added RNA-seq data of MLL-rearranged leukemia cell lines in Figure 5—figure supplement 2A and publicly available MILE study data in Figure 5—figure supplement 2B. MLL- rearranged leukemia cells (i.e., HB1119, MV4-11, RS4-11, ML-2, THP1, and EOL-1) expressed HOXA9 and MYC. Although *MEIS1* is a well-known MLL target gene, some MLL-rearranged leukemia cell lines did not express *MEIS1* (Figure 5—figure supplement 2A), consistent with the data shown in Figure 4A, B. *BCL2* and *SOX4* are expressed in all the MLL- rearranged leukemia cell lines tested (Figure 5—figure supplement 2B). However, their expression levels are not consistent among the cell lines. Analysis of publicly available data (i.e., the MILE study) showed most of the HOXA9 high leukemia samples expressed SOX4 and BCL2 at high levels (Figure 5—figure supplement 2B). *IGF1* was expressed only in MV4-11 cells among the MLL- rearranged leukemia cell lines tested (Figure 5—figure supplement 2A). These notions support our conclusion that HOXA9 is a transcription maintenance factor, but not a transcription initiation factor. The expression of HOXA9 does not initiate the expression of all the HOXA9-target genes. It perhaps maintains the expression profile of the cell-of-origin where MLL gene rearrangement occurred. Some HOXA9-target genes such as *BCL2* and *SOX4* promote leukemogenesis, therefore their expression tends to be maintained at high levels. Pan et al. showed that BCL2 was highly expressed in MLL-rearranged leukemia patients and many MLL- rearranged leukemia cell lines such as MOLM-13 and THP1 were sensitive to BCL2 inhibitor (Pan et al., 2014). High SOX4 expression was also shown to be correlated to poor prognosis of AML(Lu et al., 2017). Those studies were mentioned in the discussion in the revised manuscript.

7. The authors showed that Myc expression is comparable between HOXA9-IC and MLL-ENL-IC (Figure 1C), indicating that both HOXA9 and MLL-ENL could support Myc expression as is also argued by authors. This information is inconsistent with the results that co-expression of HOXA9 and MYC induces leukemia whereas HOXA9 single expression does not (Figure 4E). The authors should explain this apparent discrepancy.

In MLL-ENL or MLL-AF10-ICs, endogenous *Myc* is roughly 20% more expressed at the RNA levels compared to HOXA9-ICs (Figure 1C). MYC proteins are also roughly ~20% more expressed in MLL-AF10-ICs compared to HOXA9-ICs (Figure 3A). It is unclear why such a “not-so-drastic” difference would make big differences in oncogenesis as the reviewer pointed out. As shown in (Figure 4—figure supplement 2 and Figure 5—figure supplement 1A), endogenous *Myc* expression is decreased if the human MYC transgene is overexpressed, suggesting that endogenous *Myc* expression is downregulated by the excess amount of MYC or a population with low endogenous *Myc* expression preferentially survived upon selection of MYC-transduced cells. This suggests that there is an upper threshold of MYC levels that a cell can persevere as excess MYC proteins make hematopoietic progenitors prone to apoptosis. As a result, the most proliferative cells would express MYC as much as possible yet not at an exceeding level that induce apoptosis. We speculate that this 20% increase of MYC levels makes a big difference in leukemogenic capacity. HOXA9 alone likely cannot maintain Myc expression at such a high level, which may be the reason why it cannot induce leukemia in vivo efficiently. Hence additional MYC expression to HOXA9-transduced cells confers leukemogenic ability in vivo. We mentioned this in the discussion of the revised manuscript.

8. Moreover, Myc silencing at ~50% expression level of control completely abrogated colony formation in MLL-ENL-IC and HOXA9-IC (Figure 1E). Is there fine-tuning system that requires critical expression level of Myc? Or did the authors obtain much more silencing effect at the protein level? Please clarify this point.

As mentioned above, we speculate that there is a fine-tuning system that regulates Myc expression. However in this case, we think this is due to the nature of the experiment. When a gene critical for proliferation such as Myc is knocked down, cells with higher knockdown efficiency will be quickly removed from the population. Consequently, the viable cells remaining after selection tend to show relatively mild knockdown like ~50%. We provided western blot data of MYC expression in cells transduced with shRNA against Myc in Figure 1—figure supplement 2 to support the knockdown at protein levels. We saw similar phenomenon in the knockdown experiment of MENIN, an essential cofactor of MLL fusions (Yokoyama and Cleary, 2008). Hence, we think that the reason why we see only ~50% knock down of MYC is because MYC-depleted cells are quickly depleted and therefore the expression profile reflects on the cells with relatively mild knockdown.

9. In relation to the above questions, given that Myc expression is upregulated by HOXA9 like by MLL-AF10, there should be common genetic pathways regulated both HOXA9 and MYC. I would request the authors to provide the gene list and pathways commonly regulated by HOXA9 and MYC in Figure 2.

To identify the pathways upregulated by HOXA9 and MYC, we performed RNA-seq analysis of cKit+ cells and compared to those of HOXA9-ICs and MYC-ICs in Figure 2—figure supplement 1B 2A, B. Genes involved in lipid metabolism appeared to be the pathways commonly regulated by HOXA9 and MYC.

10. In discussion (page 21, lines 6-8) the authors claim that MLL fusion proteins promote oncogenesis by activating both MYC- and HOXA9-related pathways. However, the cooperative effect of MYC for HOXA9 is much weaker than that of MEIS1 as is shown in Figure 4E. Given that MEIS1 is the direct target of MLL fusions, I wonder whether MYC is dispensable in the presence of MEIS1. Since there have been several studies on MEIS1 function in leukemogenesis, the authors should show common and/or distinct downstream genes/pathways between MYC- and MEIS1-driven leukemogenesis.

MYC knockdown of HOXA9/MEIS1 cells resulted in a drastic attenuation of proliferation (Figure 4—figure supplement 1B), indicating that HOXA9/MEIS1 leukemia cells critically require MYC expression. Thus, HOXA9/MEIS1-mediated leukemia uses the MYC-dependent pathway as MYC/HOXA9-meidated leukemia. Because MEIS1 is known to associate with HOXA9, it likely enhances HOXA9-meidated functions to promote leukemogenesis. RT-qPCR analysis of 3-independent clones of HOXA9/vector- and HOXA9/MEIS1-transduced cells showed that HOXA9/MEIS1-transduced cells expressed Myc nearly twice as much as HOXA9/vector-transduced cells (Figure 4—figure supplement 1A). Thus, one possible mechanism by which MEIS1 contributes to leukemic transformation is that MEIS1 increases MYC expression cooperatively with HOXA9.

However, it should be noted that MEIS1 may confer additional oncogenic ability independently of HOXA9, as the MYC/MEIS1 combination also caused leukemia in vivo (Figure 4E). Thus, MEIS1-specific functions in oncogenesis is a very interesting topic as suggested by the reviewer.

We have performed RNA-seq analysis for HOXA9-MYC- and HOXA9-MEIS1-LCs and analyzed by gene set enrichment analysis (GSEA). In HOXA9-MEIS1-expressiong cells, genes involved in Tnfα signaling or Kras pathways were uniquely enriched. The gene sets for Tnfα signaling included leukemia associated genes such as Jun and Trib1 (Yoshino et al., 2021; Zhou et al., 2017). Although this is not the focus of the present study, this comparison suggests that MEIS1-specific downstream pathways can also be the causatives of MLL-leukemia. We have added these data in Figure 4—figure supplement 4A, B and mentioned in the revised manuscript.

11. BCL2 and SOX4 knockout showed delay in MLL-AF10- and HOXA9/MEIS1-induced leukemogenesis (Figure 7A, B). Please provide the data wheter these delays are caused by increased apoptosis by presenting the Annexin V staining as well as Caspase 3 cleavage and H2AX expression.

We have analyzed apoptotic state of leukemia cells harvested from the moribund recipient mice, the data of which were described in the Figure 7—figure supplement 2B, C. While leukemia with HOXA9/MEIS1 were slightly apoptotic in the absence of Bcl2/Sox4, we did not detect substantial differences in apoptotic signals for MLL-AF10-expressing leukemic cells. These results are partly consistent with our in vivo experiments showing both Bcl2 and Sox4 KO more significantly delayed the onset of HOXA9-MEIS1-induced leukemia compared to that of MLL-AF10-induced leukemia. Nonetheless, Bcl2 and Sox4 knockout did not cause marked differences of apoptotic properties in leukemia cells compared to the wildtype controls. We reason that these full-blown leukemia cells have acquired enough anti-apoptotic functions via multiple pathways at the time of disease presentation, therefore, the differences of anti-apoptotic properties of the leukemia cells between WT and KO are less obvious.